# Start small: A model for tissue-wide planar cell polarity without morphogens

Abhisha Thayambath[1,2], Julio M. Belmonte[1,2]*

1 Department of Physics, North Carolina State University, Raleigh, North Carolina, United States of America, 2 Quantitative and Computational Developmental Biology Cluster, North Carolina State University, Raleigh, North Carolina, United States of America

* jbelmon2@ncsu.edu

## Abstract

Planar cell polarity (PCP) is an important patterning mechanism in both vertebrates and invertebrates by which cells coherently polarize along the apical surface of the epithelium. This patterning mechanism acts upstream of many developmental processes, such as oriented growth, division, cell movements and orientation of body hairs. While various models have been proposed to explain PCP patterning, all rely on persistent global cues/gradients to obtain global orientation of large tissues. However, recent experimental work has shown that this process can happen independently of such global cues, challenging the current paradigm. In this work, we developed a new model for PCP using the Cellular Potts modelling framework to investigate the conditions under which global tissue orientation can be achieved without a tissue spanning morphogen. We found that a combination of a local boundary signal, a small initial tissue size and uniform proliferation can effectively establish long-range polarity without the need for global cues. We also investigated the impact of cell division planes and growth rates on final patterning. Finally, we compared the cell-autonomous and cell non-autonomous versions of our PCP model, as found in flies and mice, and found that the latter offers more robust patterning outcomes in the absence of gradients.

## Author summary

Cells must coordinate the spatial organization of certain proteins inside them to produce aligned structures like body hairs and cilia across large distances in various organisms. This polarity of cells is also important to provide directionality for several key events during animal development. It was widely assumed that a graded signal acting across the tissue provides the global cue for the cells to attain this long-range alignment. However, the identity and existence of this global cue have recently been challenged by new experimental findings. In this work

**Data availability statement:** The simulation code is publicly available at https://github.com/abhisha-ramesh/PCP_Subcellular_Potts_Model.

**Funding:** This work was partially funded by the Eunice Kennedy Shriver National Institute of Child Health and Human Development of the National Institutes of Health under award number R01HD117958. The funders had no role in study design, data collection and analysis, decision to publish, or preparation of the manuscript.

**Competing interests:** The authors have declared that no competing interests exist.

we developed a new computational model showing how coherent tissue polarity can emerge from simple, local rules in the absence of a global tissue-spanning signal. Our model proposes a mechanism where local signals polarize a small initial group of cells, and their subsequent proliferation extend this initial alignment across large scales. We extensively tested our model under different conditions and show that this proposed mechanism provides a reliable recipe for the generation of large-scale tissue polarity without the need for global signals.

## Introduction

Cell polarity is the intrinsic asymmetry in the shape, structure, or spatial organization of cellular components that enables cells and tissues to perform specialized functions. **Planar Cell Polarity (PCP)** refers to the coherent orientation of cells within the plane of an epithelial sheet and is a key mechanism for establishing long-range tissue organization on the apical surface. The establishment of a coherent proximal-distal axis is crucial for the proper function of many cellular and developmental processes downstream of PCP such as the generation of body hairs (sensory hairs in mammals [1,2], wing hair in *Drosophila* [3] etc.) and the orientation of tissue processes such as convergent-extension [4], gut elongation [5], neural tube closure [6,7], as well as in processes in mesenchymal tissues such as gastrulation, migration and cell differentiation [8,9].

While the mechanisms of local planar cell polarity through direct cell-cell communication are well understood experimentally and computationally, the long-range order and direction of tissue orientation remains an open question. Multiple computational models suggest the need of some global cue for the establishment of tissue wide PCP alignment in the presence of noise. Yet, experimentally, the list of possible candidates for such a global cue continues to diminish. Early in 2000's, Peter Lawrence and colleagues showed that PCP is independent of the DPP, Notch, EGF, and FGF signalling pathways [10]. The Ds and Fj proteins, which are part of the Fat–Dachsous PCP pathway, are expressed in opposing gradients and could act together as the global cue. However, uniform expression of those components does not alter the long-range PCP alignment [11]. More recent studies show that long-range PCP polarity in the *Drosophila* wing and notum can be established independent of Wnt gradients [12,13]. All these negative results suggest that an alternative mechanism, independent of morphogen gradients, may be required to establish global PCP orientation.

### PCP pathways

The mechanisms underlying planar cell polarity have been extensively studied in *Drosophila* where PCP controls the orientation of hairs on the wing and abdomen and the organization of ommatidia in the eye [14]. Two major pathways establish PCP in *Drosophila* wing- the **core PCP pathway** and the **Fat–Dachsous pathway**. Both pathways rely on the asymmetric localization of their associated proteins on the

opposing sides of each cell. This asymmetric localization is guided in part by direct communication between neighbouring cells which determines local alignment, and by a putative global cue which determines the global direction of tissue polarization.

The core pathway is composed of the cytoplasmic proteins Dishevelled(Dsh), Diego(Dg) and Prickle(Pk), transmembrane proteins Frizzled(Fz) and Van Gogh/Strabismus(Van/Stbm) and an atypical cadherin Flamingo(Fmi). Fmi is found on both opposing ends of each cell, and facilitates homophilic cell adhesion between adjacent cells [15,16]. The communication between neighbouring cells occurs through the interaction of Fz–Fmi in one cell with Vang–Fmi in the adjacent cell. Inside a cell, Dsh binds with Fz [17], whereas Pk interacts with Van [18]. Pk also binds to Dsh preventing it from localizing to the proximal side [19] by destabilizing Fz binding [20]. Diego competes with Pk for Dsh binding favouring Dsh accumulation on the distal side and thereby enhancing Fz localization at the distal membrane [21]. As a result of this competitive inhibition, the complexes Van-Pk and Fz-Dsh-Dg localize to the opposing ends of each cell, forming the cells' proximal and distal domains, respectively [22–24] (see [25] and [26] for review).

The Ft-Ds pathway is composed of the protocadherins Ft and Ds which form heterophilic complexes between cells [27]. This is regulated by the transmembrane kinase Four-Jointed (Fj). In the developing wing, the asymmetric localization of Ft to the proximal side and Ds to the distal sides of the cells is driven by the complementary gradients of Ds and Fj [28,29]. It is still unclear whether the Ft-Ds pathway operates independently or together with the core pathway to coordinate polarity across the tissue [30,31].

## Morphogens and global alignment

Experiments and mathematical models suggest that the spatial segregation of polarity complexes inside each cell and their preferential affinity between neighbouring cells is effective in coordinating polarity across small groups of cells. An initial uniform distribution of these core proteins spatially segregates over time, resulting in a polarized distribution (Fig 1A). While the reinforcement of the initial asymmetry by feedback interactions is sufficient for the local coordination of cell polarities, the identity of the biasing cue that could determine the global alignment direction and ensure large-scale coordination remains poorly understood. It has been hypothesized that the global alignment of core PCP proteins arises from a spatial gradient of a biasing cue acting across the entire tissue. Lawrence et al. have shown that if the spatial gradient of a secreted molecule acts as the global cue, then it is not part of the DPP, Notch, EGF, or FGF signalling pathways [10]. The Ft-Ds-Fj system is thought to be a potential candidate that could act as an upstream global cue for the core proteins, given that Ds and Fj are expressed in complementary gradients in opposing directions. However, even in conditions of uniform expression of these components, polarity can still be effectively established, suggesting that the gradients of Ft-Ds-Fj are not strictly necessary [11] (see [32] for a comprehensive review).

Wnt ligands, which bind to Fz receptors, are expressed as gradients in the developing Drosophila wing [33–35]. Together, they recruit and activate Dsh, which is also a highly conserved component of the Wnt signalling pathway, which in turn propagates the signal internally to the cell [36]. Since Fz and Dsh belong to the core pathway, Wnt was long considered to be the global cue that sets the global alignment and long-range orientation of PCP in *Drosophila* wing and other PCP systems. Although some studies have suggested the role of Wnt ligands (Wg and Wnt4) in global PCP establishment [37], two independent recent studies show that Wnt ligands are not required to align core PCP in *Drosophila* [12, 13]. Ewen-Campen et al. conducted loss of function studies targeting different combinations of Wg and other Wnt genes (wnt2, wnt4, wnt5, wnt6, wntD, wnt10) in *Drosophila* wing and notum and found that loss of individual or combinations of Wnt ligands that they examined do not disrupt PCP alignment [12]. In another study published in the same year, Yu et al. knocked out the diffusible Wnts DWnt2, Dwnt4, DWnt6, and DWnt10 and found no disruptions in wing hair orientation [13]. Together, these results show that Wnt ligands are not required for global PCP establishment, and other mechanisms must be responsible for the long-range orientation of PCP.

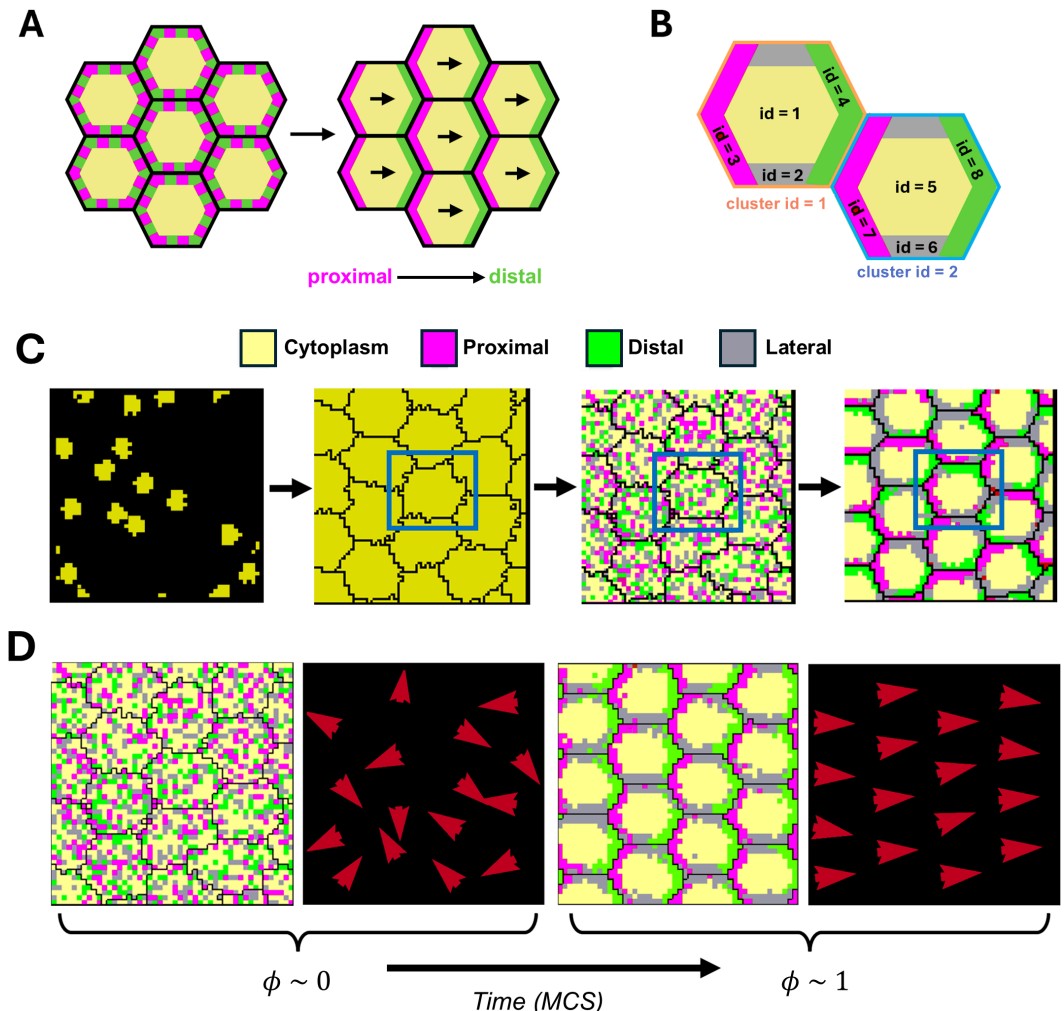

**Fig 1**. **Sub-cellular potts model for planar cell polarity: (A)** Schematic of an epithelial cell arrangement with PCP protein distributions. Initially, the PCP proteins are uniformly distributed within each cell. Over time, Fz, Dsh, and Dgo (magenta) localize to the proximal/anterior edge, while Vang and Pk (green) segregate to the distal/posterior side. **(B)** Schematic of two epithelial cells with cluster ids 1 (orange) and 2 (blue) respectively, along with the respective domain ids. The cluster id indicates the association of each domain with a specific biological cell. **(C)** Model Overview: Each epithelial cell is subdivided into four compartments—proximal, distal, lateral, and cytoplasm. **Seed** cells are placed randomly within the lattice, which grow and rearrange their shapes to efficiently pack in the space. PCP domains are introduced with random initial distributions. The proximal and distal domains spatially segregate into opposing ends of each cell over time. Blue box highlights the evolution of a single cell during this process. **(D)** The vector field representation of PCP orientations shows the transition from a homogeneous distribution ($\phi \approx 0$) of global polarization to complete alignment ($\phi \approx 1$). (Panel A adapted from [25]).

## Computational models of PCP

There are several models that explain the mechanisms behind planar cell polarity- ranging from explicit molecular signalling models to phenomenological/reduced models. The *explicit signalling models* consider the detailed interactions between the proteins in core and Ft-Ds pathways. For instance, both [38] and [39] describe cell polarity based on the asymmetric distribution of intercellular protein complexes within cells and their associated PCP components modelled by reaction-diffusion equations. Both models use a global cue (which could arise from a directional signal within each cell or

from a tissue wide ligand gradient) to introduce a small bias in each cell which are enhanced by local feedback interactions between the core proteins. Fisher and colleagues [40] have shown that both models require a persistent global cue to generate a robust polarity that is insensitive to noise.

The *phenomenological models* simplify the PCP mechanism by reducing the complexity of the spatial protein interactions to extract the fundamental interactions leading to planar polarization. Burak et al. [41] proposed a semi-phenomenological model that incorporates mutual exclusion of Fz and Vang within cells and intercellular interactions, showing that a weak early global cue can establish long-range polarity. Wang et al. [42] used a cellular automaton model where cells align their polarity with the average of their neighbours, finding that stronger global cues accelerate alignment. These models used cells with a fixed shape in a fixed lattice. The work by Zhu and colleagues [43] was the first to use cells with varying shapes and dynamic neighbourhoods, revealing that directional cues are critical for PCP, while hexagonal packing has a minimal effect on patterning. Schamberg et al. [44] developed a feedback and diffusion-based model that generates polarization in a row of cells, either through a weak initial global polarization across all cells or via a travelling wave originating from a single cell or boundary. Abley et al. [45] showed that individual cells can polarize through random fluctuations and spread polarity via cell-cell coupling and boundary organizers. This model also used gradients to improve the tissue-level polarity. Hazelwood and Hancock [46] developed a functional model using global cues to study the molecular functions of core proteins Pk and Dsh. Fisher et al. [47] explored how intercellular protein complexes interpret tissue-wide gradients when establishing polarity. More recently, Singh et al. [48] developed a minimal model centered on Ft-Ds heterodimer formation, demonstrating that while PCP can arise without a global gradient, such a gradient is essential for robustness to noise and stabilization of polarity patterns.

All these models, explicit and phenomenological, used a global cue—whether as a persistent directional signal within each cell, a tissue-wide ligand gradient, or a weak orienting signal—to establish and maintain polarity across the tissue. However, given the recent experimental results by [12] and [13] which found no evidence for the necessity of a Wnt gradient for the establishment and maintenance of PCP in the *Drosophila* wing (the primary model system for PCP), the question arises as whether there are morphogen-independent mechanisms that can explain robust establishment of long-range tissue orientation.

## A model for long-range PCP without morphogens

To address the question of how a tissue achieves large-scale PCP alignment without morphogens, we developed a phenomenological model for PCP based on the Cellular Potts (Glazier-Graner-Hogeweg) modelling framework in which we represented the asymmetrically distributed PCP proteins as distinct compartments inside cells [49,50]. We varied the contact/interface properties of these compartments to model the mutual exclusion of these proteins inside cells and differential adhesion across cells. As discussed by Lavalou and Lecuit, adhesion motifs in the transmembrane PCP proteins play an essential role in forming the inter-cellular protein complexes in such a way that when one protein is expressed at a given cell boundary, its corresponding counterpart is also recruited to the interfacial boundary of the adjacent cell [26]. In our model, the compartments representing the proximal and distal PCP proteins at adjacent cell boundaries interact with a higher adhesion strength, leading to their preferential binding. Within each cell, we can choose the properties of the cell compartments to model either cell autonomous polarization, where a single cell can segregate its PCP domains by itself, as seen in flies [51]; or cell non-autonomous polarization [52], where PCP domains require interaction with neighbouring cells to polarize, as seen in mice. Through these interactions alone, we show that cells can locally align their polarity in the absence of a directional signal. However, this mechanism alone is not sufficient to establish a long-range direction of orientation and swirling patterns arise as we increase tissue size, even in the presence of a boundary-orienting signal that polarizes the first row of cells. To explore how we can establish and maintain global alignment along a particular direction across large numbers of cells, we examine and test a mechanism first suggested by Sagner et al. [53], where a small-scale alignment generated by the boundary signal can propagate as the tissue grows. Our results show that uniform cell

proliferation coupled with a boundary signal is a robust mechanism for large-scale polarity establishment in the absence of a tissue-wide morphogen gradient.

## Model overview

We developed our model within the Cellular Potts Modelling framework, using the CompuCell3D (CC3D) simulation software [54]. This modelling framework represents cells as extended objects formed by a collection of lattice sites on a square grid in 2D or 3D. Each cell consists of either a single domain of lattice sites (defined by a unique domain ID and a domain type) or multiple domains representing specific sub-cellular compartments (e.g., nucleus and cytoplasm) or abstract regions inside cells (e.g., apical and basal regions). In our model, we represent each biological cell as a 2D object subdivided into four sub-cellular compartments/domains: (i) proximal, (ii) distal, (iii) lateral, and (iv) cytoplasm. Together they represent the spatial heterogeneity in the distribution of various PCP proteins as well as the cytosolic constituents of the cell as viewed from their apical side (Fig 1B and 1C). Since each biological cell in our model is made up of the four domains, we assign a shared cluster ID to each domain to clearly indicate its association with a specific biological cell (Fig 1B). Proximal and distal domains represent the proteins that localize to the proximal (Fz, Dsh, Dgo) and distal (Van/Stbm, Pk) regions of the cells whereas the lateral domain represents the regions of the cell membrane not populated by the core PCP proteins.

An effective energy equation $H_{total}$ defines the properties of each of these domains, such as size, aspect ratios, adhesion preferences, and interactions with other domains (Eq 1). Each of these properties is governed by an individual term in the overall effective energy equation: a volume constraint $H_V$ helps to maintain the size and compressibility of cell domains and a contact/adhesion energy ($H_C$) specifies the relative strength of contact/adhesion between domains from the same and/or different cells.

$$H_{total} = H_V + H_C \tag{1}$$

The system evolves by minimizing the effective energy $H_{total}$ by a series of random lattice-site copy attempts where we randomly select a lattice site $i$ and a neighbouring lattice site $j$ within the 3rd neighbour order of $i$. If the two lattice sites belong to different cells or cell domains, we evaluate the difference in energy ($\Delta H$) if the ID of site $i$ is copied over site $j$. Lattice site copy attempts that decrease the effective energy of the system are accepted, whereas those that increase the effective energy of the system are accepted with a probability:

$$P = \exp(-\frac{\Delta H}{T}) \tag{2}$$

where $T$ is a parameter that specifies the level of noise/membrane fluctuations in the system. The unit of time of the simulations is the Monte-Carlo step (MCS), defined as $N$ lattice-site copy attempts, where $N$ is the number of lattice sites in the grid.

The volume constraint ($H_V$) penalizes the deviations of the cell domain's volume ($V$) - defined as the number of lattice sites occupied by the cell domain - from its target volume ($V_t$):

$$H_V = \lambda_V(V - V_t)^2 \tag{3}$$

where $\lambda_V$ represents the strength of the volume constraint and can be interpreted as the inverse compressibility of the cell domain. The parameters used for the implementation of all these constraints are given in S1 Table.

The contact energy ($H_C$) is modelled as:

$$H_C = \sum_i \sum_{j \neq i} J(\sigma_i, \sigma_j) \qquad (4)$$

where the second sum is over the 4th order neighbours of site $i$; $\sigma_i$ and $\sigma_j$ are the types of cell compartments occupying sites $i$ and $j$, respectively; and $J$ is a matrix containing the contact/adhesion energy between compartment types belonging to the same cell (denoted as $j_{xy}$) and different cells (denoted as $J_{xy}$). For the sake of simplicity, we assume no variability in cells' contact energies between cells and over time.

The value of the $J$ matrix is different depending on whether the domains belong to the same cell (same cluster ID) or to different cells (different cluster IDs). When belonging to the same cell, all contact energies of pairwise combinations of domains are set at low values, while those between the proximal and distal domains are set at high values to facilitate the sorting of the two domains inside the cells (Table 1). Even in the absence of a neighbour or an interacting environment, the proximal and distal domains segregate to opposing ends inside a cell in relatively faster time scales (S1A Fig), and hence individual cells are capable of establishing PCP asymmetry autonomously in this model. This is consistent with the findings of [51] where individual cells lacking the core protein Fmi — and thus are unable to communicate with neighbouring cells — still generate PCP asymmetry. The time scale at which individual cells establish polarity is approximately 20 MCS (S1A Fig) and this is defined as the *polarization time*($\tau$) in our model (S1A Fig). We can also model cell non-autonomous systems, such as seen in mice [52], by setting all internal contact energies to negative values. In this model, the cell domains remain mixed inside the cells in the absence of a neighbour or interacting environment.

When belonging to different cells, all contact energies of pair-wise combinations of domains were set to higher values. We defined the external contact energies in such a way that proximal and distal domains of neighbouring cells would preferentially bind together to incorporate the observed inter-cellular coupling (Table 2). We determined the exact values for each contact energy term by extensive parameter scans aimed at optimizing cell autonomous polarization for small systems (S2A, S2B and S2C Fig). The parameter scans show that our results are valid for a wide range of parameter values.

We also used a temporary cell type, called **seed** cells, to initialize the simulations by randomly placing them in the lattice and allowing them to grow into arbitrary shapes until they reach a volume that completely fills the space. During this growth phase, cells remodel their shapes and rearrange their neighbours to achieve efficient packing within the lattice. This mechanism is consistent with the observation that epithelial cells are usually irregularly packed, as in the case of the

**Table 1**. *Internal* contact energies (*j*) between PCP domains.

| Cell Type | Proximal | Distal | Lateral | Cyto |
|---|---|---|---|---|
| Proximal | - | 30 | 1 | 1 |
| Distal | - | - | 1 | 1 |
| Lateral | - | - | - | 1 |
| Cyto | - | - | - | - |

**Table 2**. *External* contact energies (*J*) between PCP domains and medium.

| Cell Type | Proximal | Distal | Lateral | Cyto | Medium |
|---|---|---|---|---|---|
| Proximal | 12 | 7 | 12 | 30 | 10 |
| Distal | - | 12 | 12 | 30 | 10 |
| Lateral | - | - | 14 | 30 | 14 |
| Cyto | - | - | - | 50 | 30 |
| Medium | - | - | - | - | 0 |

*Drosophila* wing during larval and pre-pupal stages, where repacking into a hexagonal array occurs only after polarization and tissue shear [55]. Once the cells reach their target sizes and relax their boundaries, we establish PCP domains within each cell by assigning them random spatial distributions within each cell (50% of cell volume is composed of cytoplasmic domains, 15% is composed of proximal domains and distal domains each and the remaining 20% is composed of the lateral domains). This random assignment is done to avoid imposing any pre-pattern on the tissue. The total volume of each domain inside each cell remains conserved throughout the simulation, unless there is cell growth.

Starting from a homogeneous distribution, differential contact preferences within individual cell domains and between neighbouring cells lead to the spatial segregation of PCP proteins inside each cell (Fig 1C and 1D). We observe that this intercellular coupling through stronger adhesion between proximal and distal domains across cell boundaries is sufficient for local coordination of polarity in the absence of a directional cue. The proximal and distal domains segregate to opposite ends of each cell, with lateral compartment symmetrically localizing between them. This rearrangement occurs naturally, as the internal contact energies are set to prevent the direct contact between proximal and distal domains with the lateral compartment acting as a buffer between them. The system globally aligns along a random direction on a relatively short timescale of 30,000 MCS (or $1,500\tau$) for a system of $4 \times 4$ cells (S1B Fig).

When implementing cell growth and division, we update the target volumes of all cell domains such that the proportions of proximal, distal and lateral domains remain the same with respect to the perimeter of the cell. The rate of cell growth is a varied parameter in our model. Once the cell doubles volume, a division plane is chosen to be either along the major or minor axis of the cell. After division, the two daughter cells inherit equal amounts of all domains. These domains, however, are randomly mixed inside the new cells. Spatial segregation of the internal domains will happen according to the internal and external contact energies and the local neighbourhood of the new cells. This is done to avoid any bias and to reflect the biology of dividing cells, that lose their polarity information upon division [56]. After cells divide, they experience a refractory time $t_r$ before restarting their growth.

In order to quantify the degree of tissue polarization, we adapted the order parameter used by the flocking model community to describe the collective movement of self-propelled particles [57,58]. Here, we describe every cell in the lattice using a polarity vector ($\vec{v}(\sigma)$), defined as the normalized distance between the center of mass of **proximal** and **distal** domains:

$$\vec{v}(\sigma) = \frac{\vec{r}(\sigma_{prox}) - \vec{r}(\sigma_{dist})}{|\vec{r}(\sigma_{prox}) - \vec{r}(\sigma_{dist})|} \tag{5}$$

where $\vec{r}(\sigma_{\text{prox}})$ and $\vec{r}(\sigma_{\text{dist}})$ are the positions of the center of mass of the **proximal** and **distal** domains of cell $\sigma$, respectively. From this vector, we define the scalar order parameter $\phi$ that describes the degree of global tissue polarization as

$$\phi = \frac{1}{N}|\sum_{\sigma} \vec{v}(\sigma)| \tag{6}$$

where $N$ is the total number of cells in the lattice and $\phi \in [0, 1]$, where 0 means no alignment, and 1 means perfect alignment of PCP vectors (Fig 1D).

We also quantified the local polarity order $\phi(r)$ by measuring the degree of polarization as a function of either the distance $r$ between cells or as a function of the distance $r$ from a signalling boundary. The first version of the metric is calculated as:

$$\phi_R(r) = \frac{1}{N}\sum_{\sigma} \frac{1}{N_{d(\sigma,\sigma')<r}}\left|\sum_{\sigma'}^{d(\sigma,\sigma')<r} \vec{v}(\sigma')\right| \tag{7}$$

Here, $\vec{v}(\sigma')$ is the polarity vector of cell $\sigma'$, and $d(\sigma, \sigma')$ denotes the distance between cells $\sigma$ and $\sigma'$. For each reference cell $\sigma$, the local average alignment is computed over all neighbouring cells $\sigma'$ within a distance $r$ from it and normalized by their number $N_{d(\sigma,\sigma')<r}$. The outer average over all cells yields $\phi_R(r)$, which describes how local polarity decay with distance.

To quantify the local polarity order as a function of distance from the left boundary, we divided the tissue along the $x$-axis into bins of width equal to one cell diameter. For each bin centered at distance $r$ from the boundary, we calculated the mean polarity of all cells whose centers are less than $r$. The local polarity order as a function of the distance from the boundary is then calculated as

$$\phi_B(r) = \frac{1}{N_{d(b,\sigma)<r}} \left| \sum_{\sigma}^{d(b,\sigma)<r} \vec{v}(\sigma) \right| \tag{8}$$

where $d(b, \sigma)$ is the distance between boundary and cell $\sigma$ and the sum is done for all cells that lie within a distance $r$ from the left boundary.

## Results

Based on the modelling framework described in the previous section, we began by studying the behaviour of the model under various scenarios and checking if we can reproduce the previously known experimental results. We first examined the time evolution of the degree of global polarization across different boundary conditions. Then, we tried to reproduce the domineering non-autonomy effects in polarity alignment observed in *Vang* and *Fz* loss of function mutants. After this, we studied the impact of increasing system size under different boundary conditions and investigated the mechanisms by which cells can establish long-range polarity. All results presented are averages over 100 independent simulations.

### Time evolution of global polarization under different boundary conditions

We first characterized our model by investigating how it behaves under different conditions. We examined the evolution of global polarization over time under different boundary conditions for a system of $8 \times 8$ cells (Fig 2A and 2B). We initialized the system by randomly placing seed cells in the simulation space and letting them grow and relax their boundaries for $10^4$ MCS. After that we added the PCP domains to the cells in a random manner (Fig 1C) and studied the evolution of global polarization defined by the scalar order parameter $\phi$ (Eq 6 and Fig 2D). As time increases, global polarization increases and then stabilizes after $5 \times 10^5$ MCS (or $2.5 \times 10^4$ $\tau$)for both boundary conditions (Figs 2D and S3A).

To circumvent the effects that an external boundary might have on tissue orientation, we first simulated our model with periodic boundary conditions to mimic a continuous and unbounded tissue (Fig 2A, first panel and S1 Video). These boundary conditions help eliminate the edge effects present in large biological tissues by treating the tissue as if it wraps around on itself. Starting from an initial state where cell polarization vectors are randomly oriented (Figs 1D, first panel and 2C, first panel, green), the system self-organizes towards a globally aligned state based only on the local contact preferences between the domains within the cells and between neighbouring cells without any imposed external cues. Over time, the global polarization increases and stabilizes at approximately 85% (Fig 2D, red). The final tissue angles of polarization are predominantly distributed along four distinct directions—right, left, up, and down—corresponding to the symmetry axes of the lattice (Fig 2A, second panel). Fig 2C (second panel) illustrates one such outcome in a small $8 \times 8$ system, where the initially random orientations of cell polarity vectors (Fig 2C, first panel) evolve toward the left, representing one of the possible final states under periodic boundary conditions.

We also investigated the effect of boundary-orienting signals on tissue polarization. We added a fixed column of **distal** domain type at the left boundary of the system (Fig 2B), which acts as a directional cue that orients cells directly in contact

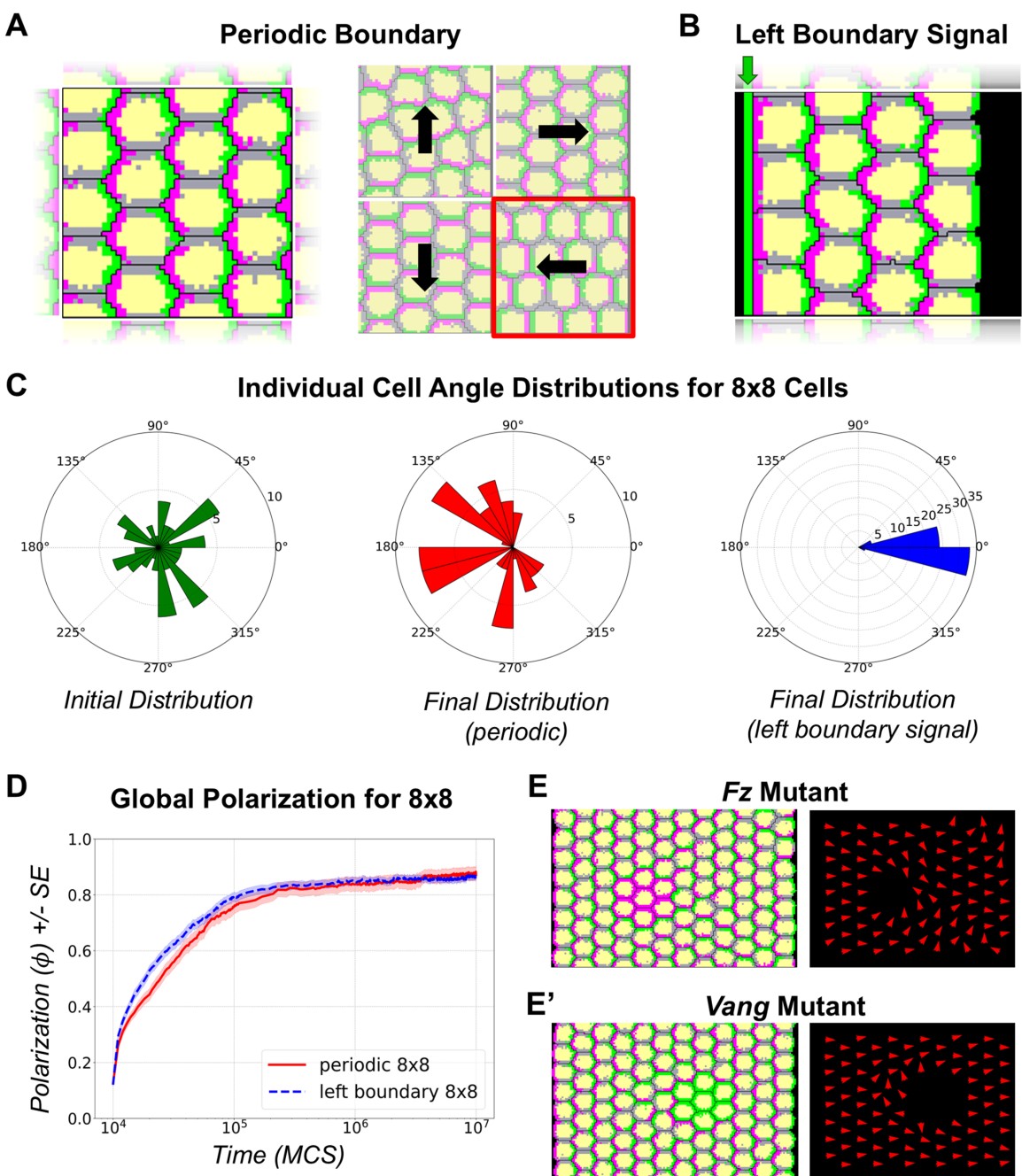

**Fig 2.** **Time evolution of global polarization under different boundary conditions and domineering non-autonomy: (A)** Polarization outcomes under periodic boundary conditions applied along both the *x* and *y* directions. The system exhibits four distinct final states, with the mean tissue polarization vector aligning along one of the four directions: right, left, up, or down. **(B)** Polarization under a directional cue applied at the left boundary, with periodic boundary conditions maintained along the *y* axis. This configuration introduces a bias to the right and sets a definitive proximal-distal axis to the right. **(C)** Distribution of individual cell polarization angles in a system comprising $8 \times 8$ cells. The first panel (green) shows the initial distribution of cell polarization vectors. The second panel (red) depicts the final distribution under periodic boundary conditions for a system that polarizes towards the left direction, similar to the output highlighted in red in panel **(A)**. The third panel (blue) shows the final distribution under the left-boundary orienting signal, with polarization vectors highly aligned along the proximal-distal axis. **(D)** Time evolution of global polarization $\phi$ under periodic (red) and left-boundary (blue dashed) conditions in an $8 \times 8$ system over $10^7$ MCS; shaded regions indicate the standard error (SE). Global polarization increases with increasing time. **(E)** Domineering non-autonomous behaviour of cells in loss-of-function mutants for *Fz*. In *Fz* loss-of-function mutants (magenta), polarity vectors distal to the mutant patch point towards the mutant clone. **(E')** Domineering non-autonomous behaviour of cells in loss-of-function mutants for *Vang*. In *Vang* loss-of-function mutants (green), polarity vectors proximal to the mutant patch point away from the mutant clone.

with it towards the right (Fig 2C third panel, blue and S3 Video). The open boundary on the front does not have a preference for proximal or distal domains. The addition of this boundary signal biases the global alignment to be predominantly directed away from the boundary, as it can be seen in the radial histogram of individual cell angle distributions (Fig 2C, third panel, blue). We also found that the presence of this boundary slightly speeds and enhances the global polarization of the tissue, as the evolution of the order parameter $\phi$ is higher than the periodic boundary configuration for nearly all time steps in the simulation (Fig 2D). From these results, we conclude that the presence of the boundary signal in our model successfully biases small systems to polarize along a preferred direction, as observed in other models for PCP [41,42].

We also changed the domain type/contact energy preferences of the left boundary to lateral domain type to see how the system polarizes. We found that this led to independent columns of cells that polarized either up or down (two possible outcomes shown in S4A Fig). As a result, the global polarization increases to only about 50% and the mean angle of tissue polarization is biased around $\pm 90°$ (S4B and S4C Fig).

### Loss of function mutants produce domineering non-autonomy

Clones of cells lacking *Fz* function disrupt polarity in the wild-type cells distal to the mutant clones, whereas clones lacking *Vang* function disrupt polarity in wild type cells on the proximal side of the mutant clone [38]. This effect is called "domineering non-autonomy" because the mutant cells affect the polarity of the neighbouring wild-type cells. To test whether our model reproduces this result, we created an $8 \times 12$ simulation space with an initial, transient left boundary signal to induce global polarization to the right, and introduced a patch of cells that contain only proximal or distal surface domains, mimicking the loss of function for *Fz* and *Vang* activity, respectively. We found that the polarity vectors of cells distal to the *Fz*-like mutant patch reoriented facing the mutant clone (Fig 2E). Conversely, we found that the polarity vectors in wild-type cells proximal to the *Vang*-like mutant patch pointed away from the clone (Fig 2E'). Hence, our model successfully reproduces the domineering non-autonomous effects of *Fz* and *Vang* mutants, as expected given that PCP alignment mechanism in our model is determined solely by local effects.

### Effect of system size on global and local polarization

After establishing the basic characteristics of the model and verifying that it is able to reproduce key results from experimental observations and other published models, we investigated the effect of system size on global and local polarization. We increased the number of cells across different lattice sizes and ran simulations for $10^6$ MCS to allow sufficient time for the systems to polarize. As in the previous section, we considered two boundary conditions: periodic boundaries and left boundary signal (Fig 2A and 2B). We found that as the number of cells increases, global polarization decreases in both configurations (Fig 3A, see S1C Fig for time series data). In the periodic boundary configuration with $30 \times 30$ cells, the distribution of the mean angle of tissue polarization is nearly uniform (Fig 3B, first panel). However, for the $30 \times 30$ system with a left boundary signal (Fig 3B, second panel) the mean angle does not exhibit the strong preference along the proximal-distal axis as observed in the $8 \times 8$ system (Fig 2C, third panel). Instead, the alignment direction is more dispersed around proximal-distal axis (Fig 3B, second panel). This suggests that the effect of the boundary orienting signal weakens with the distance from the boundary.

For the periodic boundary configuration, the representative simulation output for an $8 \times 8$ system shows that the final cell polarity vectors align along a specific direction (Fig 3C, top panel and S1 Video) while the $30 \times 30$ system shows that cells align within small local domains and swirling patterns emerge (Fig 3C, bottom panel and S2 Video). This clearly suggests that when system size increases, there is no long-range order across the entire system in the absence of a directional cue. For the configuration with the left boundary signal, $8 \times 8$ system shows cell polarity vectors aligned along proximal-distal axis (Fig 3D, top panel and S3 Video). However, when system size is increased to $30 \times 30$, although the

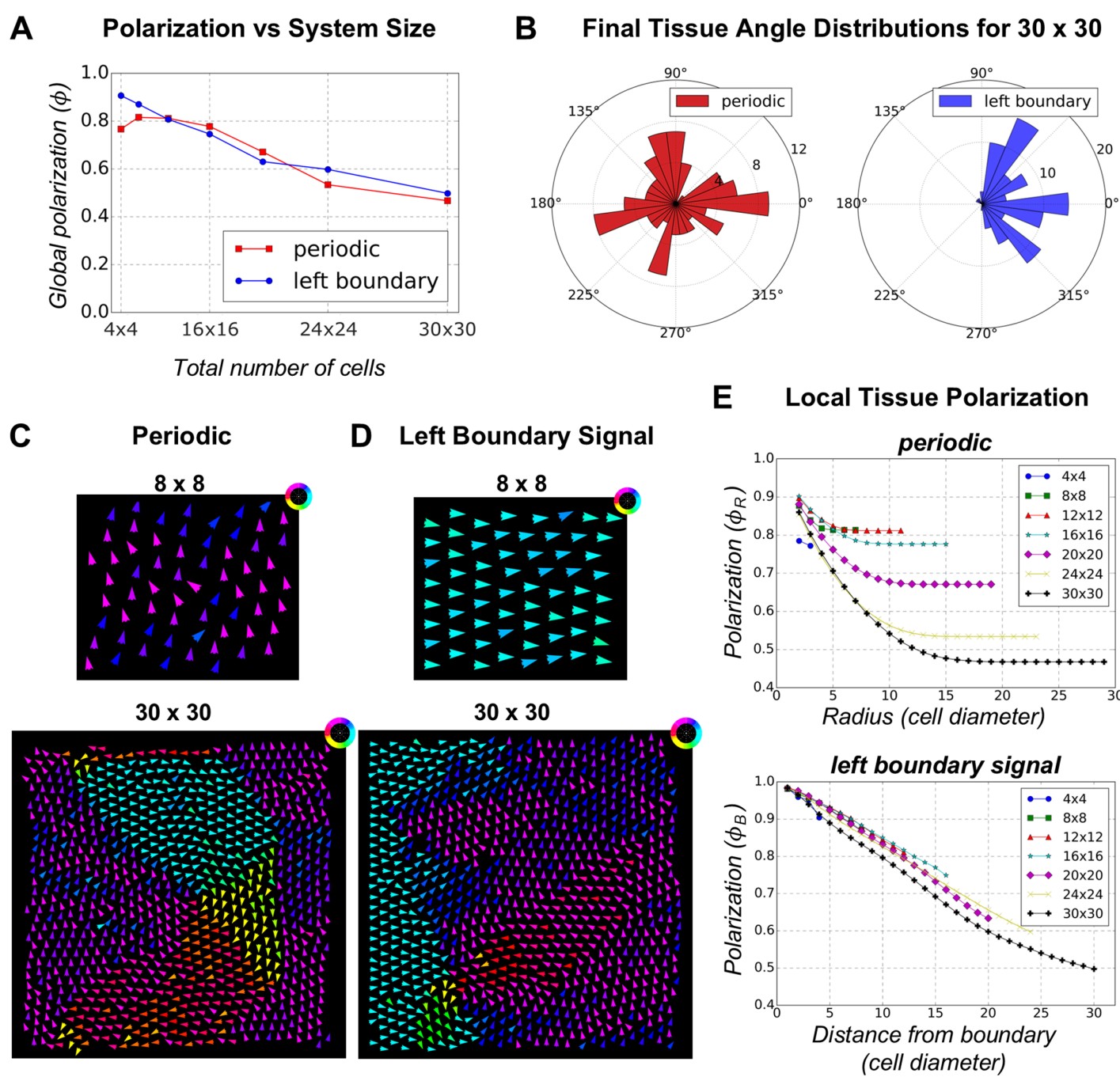

**Fig 3**. **Effect of system size on polarization for periodic boundary and left boundary signal configurations. (A)** Final average global polarization ($\phi$) for different system sizes under periodic and left boundary signal configurations (average over 100 simulations). As system size increases, global polarization decreases for both boundary conditions. **(B)** Final tissue angle distribution of polarization for the $30 \times 30$ system under periodic (red) and left boundary (blue) conditions. For periodic boundaries, the polarization angles become nearly uniform, while for the left boundary signal, the alignment direction is more dispersed around the proximal-distal axis. **(C)** Final cell polarity vectors for the $8 \times 8$ (top panel) and $30 \times 30$ (bottom panel) system with periodic boundary conditions. In the smaller $8 \times 8$ system, all cells align in a single direction (upward in this case). For the $30 \times 30$ system, local alignment occurs within small domains, and swirling patterns emerge. **(D)** Final cell polarity vectors for the $8 \times 8$ (top panel) and $30 \times 30$ (bottom panel) systems with left boundary signal configuration. For the smaller $8 \times 8$ system, all cells align along proximal-distal direction. For the $30 \times 30$ system, alignment is maintained in the first few columns but dissipates as cells move farther from the boundary. **(E)** Local polarization as a function of radial distance for periodic boundary configuration (top panel) and as a function of distance from the boundary for left boundary signal configuration (bottom). Local polarization decreases sharply with distance in the periodic boundary case, whereas the decrease in polarization for the left boundary signal configuration is less steep.

first few columns of cells remain well aligned along proximal-distal axis, global alignment is lost as we move further away from the left boundary (Fig 3D and S4 Video, bottom panel).

We also examined how polarization changes as a function of distance using the two versions of the local polarity order parameter $\phi(r)$. Local order is high (over 80%) over small distances (radius <5 cell diameters), but decreases steeply as the radius (measured in units of cell diameters) increases for the periodic boundary configuration independently of tissue size (Fig 3E, top panel). This suggests that the distance over which the polarity alignment is maintained, the correlation length, is very small. In the left boundary signal configuration, cells close to the boundary are highly polarized with $\phi > 90\%$ for distances up to 5 cell diameters. The degree of polarization still declines with distance from the boundary, but the decrease is less steep compared to the periodic case (Fig 3E, bottom panel). This indicates that the boundary signal helps maintain the cell polarity correlation higher and for longer distances as compared to periodic boundary configuration. However, it does not propagate tissue polarity effectively across large distances from the boundary. This reinforces the idea that additional mechanisms might be required for the coordination of polarity across a large tissue.

## Cell proliferation is sufficient to establish long-range polarity

Our findings from the previous section suggest that establishing polarity across a large tissue requires additional mechanisms. A key question is whether this can be achieved without a graded biasing signal acting across the entire tissue. We hypothesize that the cells may first establish polarity in a small domain with the help of a margin/boundary-orienting signal and then expand this domain while conserving alignment. To test this, we start with a small patch of tissue that is already polarized with the help of the left boundary-orienting signal. We then allow the cells to proliferate by gradually increasing their volume. We let these cells grow slowly while maintaining the relative proportions of the proximal, distal, lateral, and cytoplasmic domains so that the cells maintain their polarity during this process. When a cell exceeds twice its volume, it divides along its minor axis. Following division, daughter cells inherit equal fractions of the cell domains from the parent cell but lose internal polarity as we redistribute these domains uniformly within each daughter cell (Fig 4B). These cells re-polarize over time by spatially re-organizing their domains based on the polarity of the neighbouring cells. This mechanism is consistent with the findings from Devenport et al., where they describe how PCP components are equally inherited by daughter cells and how they re-establish their polarity in accordance with their neighbours in basal epidermal stem cells in mouse embryo [56].

We tested two different growth scenarios: growth at the distal end of the tissue (opposite to the boundary signal) (Fig 4C and S5 Video) and uniform tissue growth (Fig 4D and S6 Video). We begin our simulations with a $10 \times 4$ system of cells that are already polarized. There is a periodic boundary condition along the vertical direction, a fixed boundary wall of **distal** domain type at the left side of the tissue, and an open space to the right that the tissue occupies as the cells grow and divide (Fig 4A). The open boundary on the front has no preference between proximal and distal domains. We compare the results of the two scenarios to the polarization of tissues with fixed/pre-established sizes under the same boundary conditions that polarizes all at once. To allow for the comparison of growing tissues at different times, we varied the number of cell columns in the non-growing tissue.

As expected, for the simulations with fixed-sized tissues, the more cell columns we added, the smaller the global polarization, reaching $\phi = 0.3$ for the largest system size examined (Fig 4E, red line). The decrease in $\phi$ is steeper with increasing number of cells when compared to the previous case with square shaped tissues (Fig 2A first panel, 2B). Here each additional column of cells is farther away from the boundary signal, which causes a steeper decline in tissue polarization as a function of distance from boundary than the square tissue case, where cells are also added along the signalling boundary (Figs 4E and S5).

Next, we tested whether pre-polarizing a small ($10 \times 4$) tissue and allowing cells at the distal end to grow and divide would improve polarization (Fig 4C). As the number of cells increases, we observed a decrease in global polarization $\phi$ until the system size reaches 200 cells (Figs 4E and S6A). After this, $\phi$ gradually recovers and stabilizes at approximately

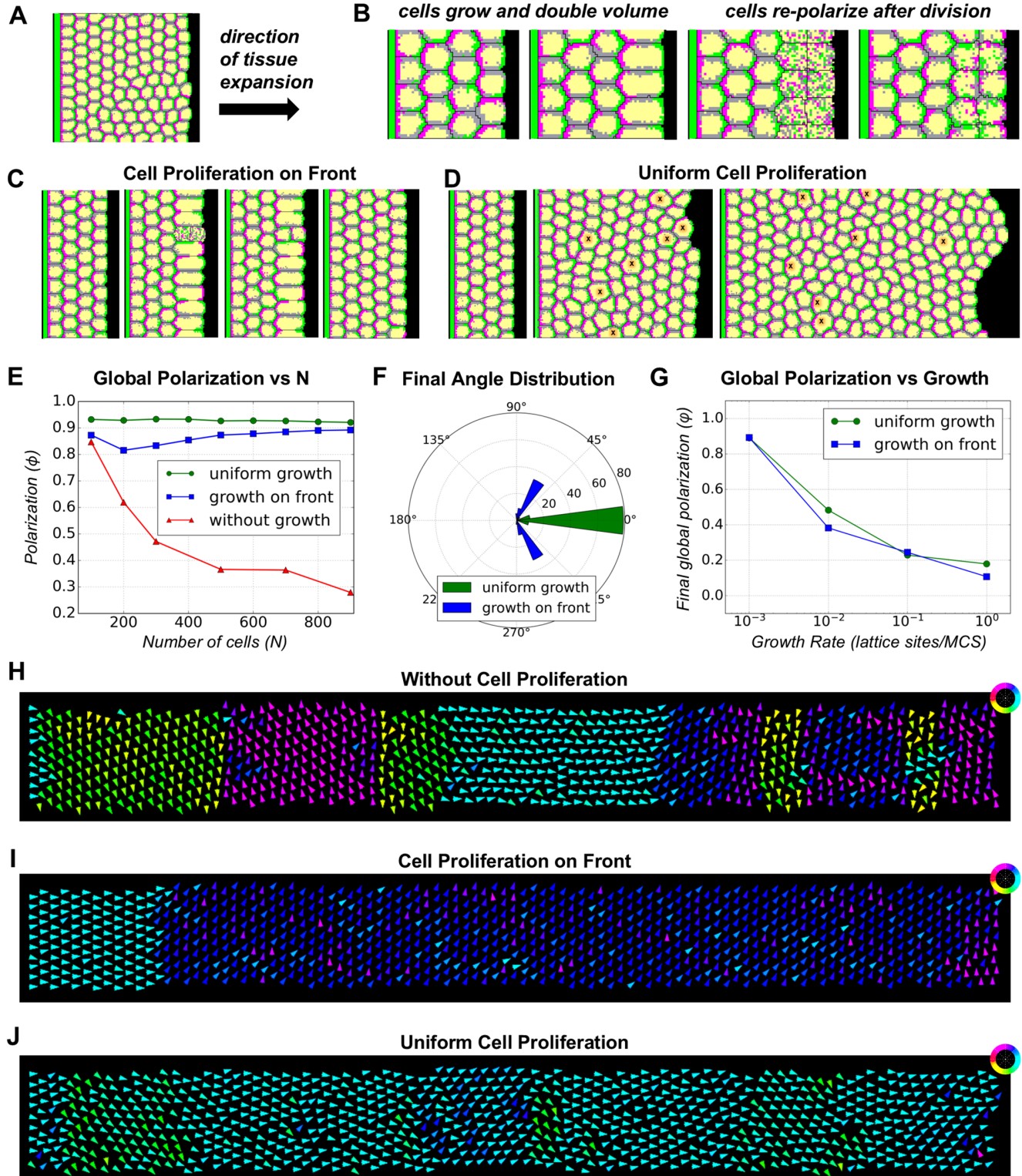

**Fig 4. Cell proliferation-based mechanism for maintaining polarity. (A)** Schematic of the system used to study the effect of cell proliferation. 10 rows of cells are arranged in a box with periodic conditions along the vertical direction, a boundary signal on the left, and an open space on the right. **(B)** Cells grow to double their volume and then divide. After each division, daughter cells inherit the PCP domains equally, lose polarity, and redistribute these domains based on neighbouring cells' polarity. **(C)** Simulation showing cell growth and division at the distal/front boundary of a system.

**(D)** Simulation showing uniform cell proliferation. Cells about to divide are indicated by an 'X' mark. **(E)** Comparison of global polarization $\phi$ for systems with and without cell proliferation as a function of the number of cells. For fixed tissue sizes, global polarization decreases with system size (red). Systems with cell proliferation on front/distal boundary stabilize at a higher $\phi$, approximately 0.89 (blue). Uniform cell proliferation further enhances global polarization at a higher value 0.92 (green). **(F)** Distribution of mean angle of tissue polarization for systems with proliferating cells, showing a shift towards $\pm 60°$ for front/distal boundary proliferation and proximal-distal alignment for uniform cell proliferation. **(G)** Comparison of final global polarization when number of cells in the system is 900 for varying growth rates for front/distal proliferation (blue) and uniform cell proliferation (green). For uniform cell proliferation, the refractory time between cell divisions is set to $10^4$ MCS. **(H)** Simulation output showing polarity vector alignment for the configuration without cell proliferation. Domains of local alignment can be observed. **(I)** Simulation output showing polarity vector alignment for the proliferation on front configuration. **(J)** Simulation output showing preference for alignment along the proximal-distal axis for the uniform cell proliferation configuration.

0.9 as system size reaches 900 cells (Fig 4E). This suggests that cell proliferation on the boundary may help maintain the initially established polarity. However, when we examine the mean tissue angle distribution for these systems, we observe that the alignment is not strictly along the x-direction but instead shifts towards $\pm 60°$ (Fig 4F). Simulation snapshots confirm that after a few rows away from the left boundary, the polarity vectors tend to point either upward or downward, explaining the initial decrease observed in global polarization $\phi$ (Figs 4I and S7). This trend can be observed in the plot of local polarization as a function of distance from the boundary, where $\phi$ decreases with distance and stabilizes after approximately 20 columns of cells (S5 Fig). The results presented are for growth rates of 0.001 lattice sites per MCS (or a $1.44 \times 10^5$ MCS volume doubling time). Results remain the same for slower growth rates, but polarization is gradually lost for faster growth rates, suggesting that cells need some time to adjust their polarity before the next round of division to maintain local tissue alignment (S8A and S8B Fig). We found that slower growth rates help maintain the initially established polarity as the cells grow (Figs 4G, S8A, S8B and S3B).

Instead of restricting cell growth to just the front boundary, we allow all cells to grow, but at randomly assigned times (Fig 4D). The time at which each cell begins to grow is decided by selecting a random number between 0 and refractory time $t_r$. Once a cell reaches twice its initial volume, it divides. The newly divided cells would start to grow again only after a random time $t_{random}$ (selected between 0 and $t_r$) has passed. The growth rate is set to 0.001 lattice sites per MCS (or a $1.44 \times 10^5$ MCS volume doubling time). As a result, the time scale for polarity adjustment is much shorter than the growth and division time scale, giving the cell ample time to adjust its polarity between divisions. We find that the global polarization, $\phi$, remains roughly constant at values greater than 0.9 as the number of cells in the system increases. Although there is a slight progressive drop in $\phi$ as the system size expands, $\phi$ stays above 0.9 up to a system size of 900 cells (Fig 4E). A similar trend is observed for the local order, which also remains above 0.9 (S5 Fig). Unlike the previous growth scenario, the mean angle distribution plot shows that the system stays preferentially aligned along the proximal-distal axis (Fig 4G). The simulation output also shows that the polarity vectors tend to point along the proximal-distal direction (Fig 4J). As with the case with front-restricted proliferation, slower growth rates and slower refractory times between divisions helped maintain the polarity as the tissue grows (S8 and S9 Figs).

To understand the reason for the difference in the two growth scenarios, we quantified the recovery of polarization by cells after division by comparing the relative time scales of polarity re-establishment in two different settings (S10 Fig). Starting from a system of cells that is already polarized, we allowed a single round of cell growth and division in a column of cells at the front boundary and inside. Cells dividing at the distal front required longer time to re-establish their polarity along proximal-distal axis compared to the cells dividing inside. This was observed independent of the plane along which cells were divided.

Next we tested whether the plane of cell division (cleavage plane) has an effect on the establishment of global polarization. We compared the polarization dynamics for both proliferation scenarios (front and uniform) under division along major (long) and minor (short) axis of the cells (S11A Fig). For uniform cell proliferation, the global polarization is maintained as the number of cells in the system increased irrespective of cleavage plane (S11B Fig). The final tissue angle

distribution also did not depend on the cells' cleavage plane and were primarily distributed along proximal-distal axis (S11D Fig). However, for cell proliferation on the front, the global polarization gradually decreased with increasing system size (about $\approx 1\%$) as the cells divided along their major axis (S11C Fig). Unlike in the case of minor axis divisions, the mean angle of tissue polarization is more dispersed around the proximal-distal axis (S11E Fig). Thus, uniform cell proliferation appears to be an effective mechanism that robustly maintains the initially established polarity.

## Cell autonomous vs cell non-autonomous polarization

The question of whether cells require contact with neighbouring cells to establish planar cell polarity has been investigated in both mouse and *Drosophila* systems, with contrasting outcomes. In the mouse epidermis, studies of chimeric embryos by Basta et al. (with dual PCP reporter cells with *Fz* and *Vang* markers and unlabelled Celsr1 mutant cells) show that intercellular complexes formed by the atypical cadherin Celsr1 at cell-cell junctions are essential for cells to sort the PCP proteins across cell junctions [52]. As a result, mouse epidermal cells that lack contact with neighbours fail to establish PCP asymmetry, indicating a cell non-autonomous mechanism. In contrast, Weiner et al. showed in the *Drosophila* wing that disrupting intercellular homodimerization of the atypical cadherin Flamingo (*Fmi*) by replacing it with *Fmi*Δcad, which cannot dimerize between cells but still supports intracellular PCP complex assembly, does not prevent cell polarization [51]. These experiments demonstrate that *Drosophila* wing cells can establish PCP cell autonomously, without requiring intercellular cadherin-mediated interactions.

To investigate how the model behaves under these two scenarios- cell autonomous versus cell non-autonomous polarization- we created a second version of our PCP model to simulate the cell non-autonomous case. While in the cell autonomous version the internal contact energies were all positive ($j_{pd} = 30$ and all other internal energies were set to 1), for the cell non-autonomous version, all internal contact energies were set to be negative (Fig 5A). To check whether the two versions lead to consistent domain segregation or mixing, we tracked the distance between proximal–distal compartments and their common interface over time within a single cell (Fig 5B). In the cell-autonomous version, the distance remained approximately 0.8 cell diameters, indicating that compartments remained spatially separated at all times. However in the cell non-autonomous version, the distance fluctuated between 0 and 0.8 cell diameters, showing that the compartments failed to spatially segregate (Fig 5B left panel). Conversely, the interface between proximal-distal compartments remained near zero in the cell-autonomous case, whereas it fluctuated at non-zero values in the cell non-autonomous case (Fig 5B right panel).

Next, we investigated the effect of system size on cell non-autonomous polarization under different boundary conditions. We found that global polarization decreases with increasing system size (Fig 5C). However, for both boundary conditions this decrease is less pronounced compared to the decrease observed for cell autonomous polarization (Figs 3A, S12A and S12B). The final mean angle of tissue polarization for a $30 \times 30$ cell system under periodic boundary conditions were uniformly distributed in $[0, 360°]$ whereas under the left boundary condition, the angles were more dispersed along the proximal–distal axis (Fig 5D). Simulation snapshots show that periodic boundary conditions produced small, locally aligned domains (Fig 5E, left), whereas left-boundary conditions maintained alignment along the proximal–distal axis near the boundary but lost alignment farther away (Fig 5E, right).

We also examined the time taken by the system to reach $50\%$ global polarization ($\phi = 0.5$) for these boundary conditions for cell autonomous and cell non-autonomous polarization. Systems with periodic boundaries polarized more slowly, while those with left-boundary signals polarized faster (S13A Fig, solid line). Across 100 simulations, most systems reached $50\%$ polarization even with increasing size, but compared to the non-autonomous case, cell-autonomous polarization required nearly an order of magnitude more time, and the fraction of simulations achieving $50\%$ polarization dropped sharply with size (S13A Fig, left, dashed line). When we increased system size along the x-axis while keeping 10 cells along the y-axis, the time to reach $50\%$ polarization grew with size in both cases, but much more sharply in the

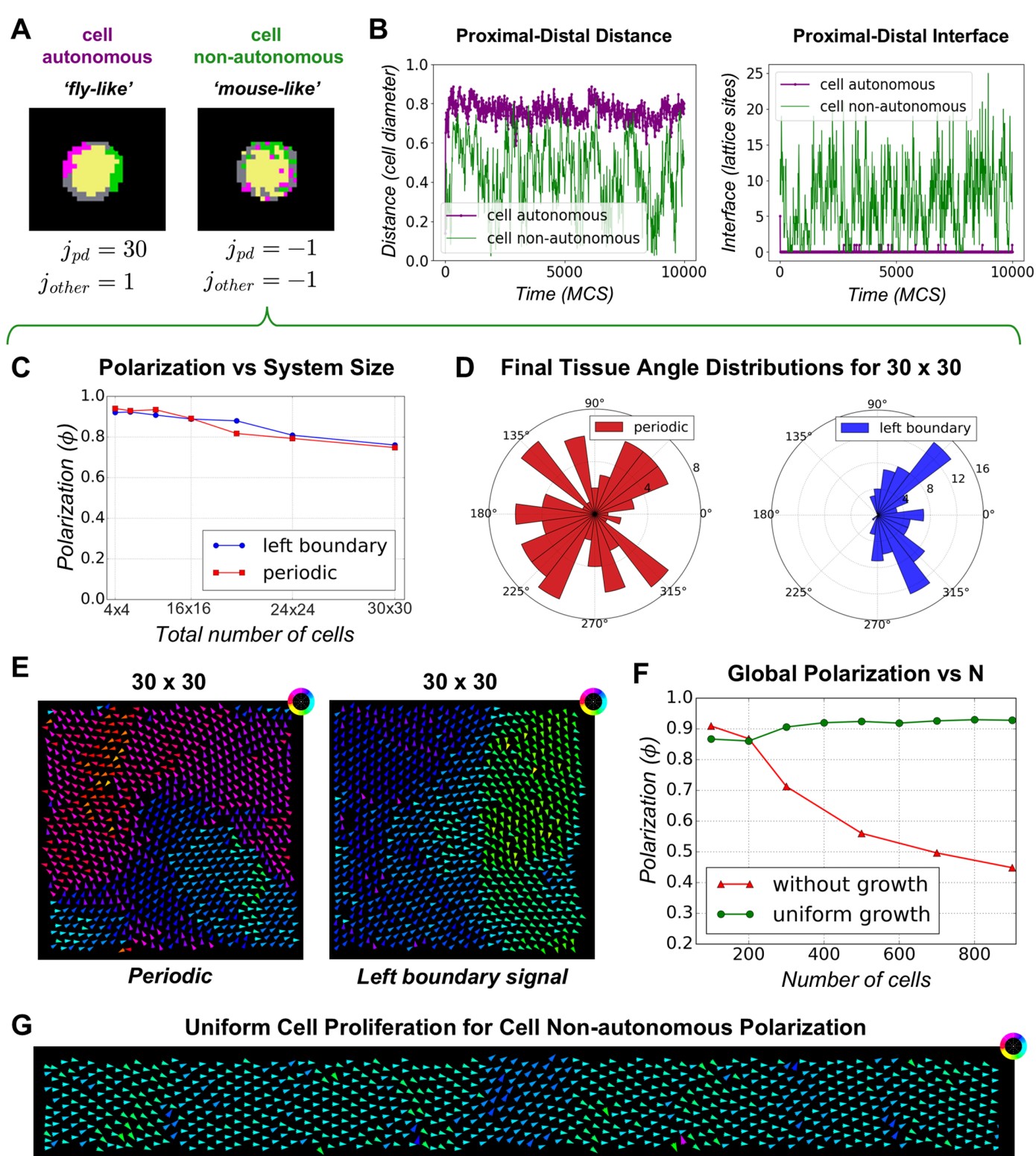

**Fig 5. Cell non-autonomous polarization. (A)** Simulation snapshots showing cell-autonomous (as observed in *Drosophila* wing) and cell non-autonomous polarization (as observed in mouse epidermis), where the PCP proteins can either self-polarize or not in single cells. **(B)** Comparison of the distance (left panel) and interface (right panel) between proximal and distal compartments over time for both model versions. In the cell-autonomous case, the compartments have no common interface and remain spatially separated, while in the cell non-autonomous case the interface area is high

and their distance fluctuates at values lower than 0.8, indicating mixing. **(C)** Global polarization of cell non-autonomous PCP model as a function of system size for periodic and left boundary signal configurations. As system size increases, global polarization decreases under both boundary conditions. **(D)** Final mean angle of tissue polarization for a $30 \times 30$ cell system: under periodic boundary conditions (left, red), the angles are uniformly distributed between 0° and 360°; under the left boundary condition (right, blue), the angles are more dispersed along the proximal–distal axis. **(E)** Final cell polarity vectors for a $30 \times 30$ system with periodic boundary conditions (left): local alignment occurs within small domains. For the left boundary signal configuration (right), alignment is maintained in the first few columns but dissipates with distance from the boundary. **(F)** Comparison of global polarization for systems with and without cell proliferation as a function of the number of cells. For fixed tissue sizes, global polarization decreases with system size (red). Uniform cell proliferation enhances global polarization at a higher value 0.93 (green). **(G)** Simulation output showing preference for alignment along the proximal-distal axis for the uniform cell proliferation configuration.

autonomous case (S13 Fig, solid line). The fraction of simulations reaching 50% polarization decreased only slightly for non-autonomous cells but fell to 20% for autonomous cells (S13B Fig, dashed line).

As in the case of cell autonomous polarization (Fig 4E), uniform cell proliferation rescued the decline in global polarization observed for fixed tissues with increasing system size (Fig 5F). Uniform proliferation enhanced and stabilized global polarization at a higher mean value (0.93) compared to the cell autonomous case (S12C Fig). Simulation outputs show a preference for alignment along the proximal-distal axis for the uniform cell proliferation configuration (Fig 5G).

## Emergence vs maintenance of PCP

An equally important question is how well a pre-established PCP order can be maintained over time. To understand this, we created perfectly aligned tissues by pre-assigning proximal and distal compartments to proximal and distal sides of each cell (after letting the seed cells relax their boundaries and pack in the lattice) and added the left boundary signal. The global polarization decreased linearly over time in all simulations independently of system size (S14B Fig). We observed that the decrease in polarization for systems with pre-established perfect PCP order is slower than the polarization of tissues of the same size (S14B Fig). Local order from the boundary also decreased with distance from boundary for increasing system size with steeper decrease in larger systems (S14C Fig). The initial and final tissue angle distributions also show that the alignment direction became more dispersed along the proximal-distal axis over time (S14A and S14D Fig). However, when we pre-established PCP in a perfectly hexagonally packed tissue, the orientation was maintained indicating that the packing geometry of the lattice has an effect in maintaining the established polarity (S15A and S15B Fig).

## Other mechanisms for PCP alignment without morphogens

We examined other alternative mechanisms that could establish long range PCP without morphogens. The first one is an extension of the mechanism that we proposed with the left boundary signal. In addition to the left boundary signal, we introduced a second signalling boundary on the distal side of the tissue, generating a double boundary signal configuration (S16A Fig). The presence of complementary boundaries on both sides of the tissue consistently increased the global polarization of cells across all system sizes when compared to the single boundary configuration (S16B and S16C Fig). Even though the global polarization decreased with increasing system size, this configuration extended the maximum tissue sizes that could have coherent alignment.

Inspired by the PCP polarization process in zebrafish retinal cone cells [59], we also explored a mechanism where cell rows are sequentially activated for PCP signalling from a local boundary (S16D Fig). In this case, each column of cells polarizes only after the preceding column has already established and stabilized its orientation. This temporally staggered activation of PCP domains in neighbouring cells could be an alternative mechanism to establish PCP in the absence of morphogens.

## Discussion

Planar Cell Polarity signalling is a very common process used by multiple organisms to pattern both mesenchymal and epithelial tissues. Given its relevance, much effort has been put by both experimentalists and theoreticians to understand how this process takes place. All models of PCP to date require some sort of global cue to ensure robust tissue alignment at large scales, a result that supported the existing assumption within the field that morphogenetic gradients such as Wnts were responsible for the observed coherent cell polarization at tissue scale. Given that two recent experimental works put that assumption in check, new modelling efforts were needed to check whether long-range PCP alignment is possible without global cues and what is required to achieve such a result.

To investigate how tissues can achieve long-range polarity without the use of global morphogens, we created a stochastic phenomenological model of PCP dynamics that relies only on the contact affinities between cell compartments within and across cells. By carefully choosing our model parameters, these local interactions lead to the segregation of proximal and distal compartments within cells and alignment of proximal-distal orientation among neighbouring cells (Fig 1C). Like all other models to date, we assume that the PCP proteins that make the distal and proximal domains have the ability to reorganize their spatial orientation using the local information provided by their immediate neighbours. This assumption was recently demonstrated experimentally by the work of the Devenport and Axelrod groups for both mice and flies, respectively [51,52]. We systematically increased the size of our simulated tissues and found that global ordering is quickly lost with the number of simulated cells, confirming that local interactions alone cannot guarantee global ordering, a result shared with many other PCP models [41,42].

Next we tested whether the presence of a local signalling boundary would be sufficient to orient the whole tissue in the same direction. Many tissues where PCP is present exhibit such local orienting signals. For example, in the *Drosophila* wing imaginal disc, PCP is aligned with respect to multiple signalling centers such as Wingless (Wg) and Notch at the dorso-ventral boundary, Hedgehog (Hh) and Decapentaplegic (Dpp) at the anterior-posterior compartment boundary, and Dachsous (Ds) at the hinge fold [53]. Planarians also have 3 signalling centers, two at the anterior and posterior poles of the animal, and one extending along the body margins [60]. We added a signalling wall on the left side of our simulated tissue to induce the first row of cells to polarize proximal-distally and an open space on the opposite side of the tissue with no contact preference to proximal or distal cell compartments to prevent a double boundary with possible competing bias. Although the presence of a boundary signal does bias the overall tissue direction, global polarization still decreases with system size at the same rate as in the previous simulations without a boundary signal. Addition of a second boundary on the right side with proximal properties does reinforce the directional bias and increases the overall range of patterning, but does not prevent the loss of global polarization with increasing number of cells.

Given that for small systems a local boundary signal successfully leads to full tissue polarization in the direction specified by the boundary, we tested whether the combination of an initially small tissue, a local boundary and subsequent cell proliferation would suffice to generate large scale tissue orientation - a hypothesis first proposed by Sanger et al. in 2012 [53]. In our model for cell growth and division, we incorporated two important observations made in mammalian skin cells: that PCP components are inherited by daughter cells; and that cells lose their polarity information during division due to the temporary internalization of PCP proteins [56]. Therefore, cells in our model must reorient their proximal and distal compartments after division using only the local information from their immediate neighbours. Our results show that cell proliferation is sufficient to sustain and propagate the initially established polarity in an expanding tissue (Fig 4J). The ability of the tissue to sustain global polarization while growing is dependent on cell growth rates, with faster division cycles leading to an increasing loss in tissue polarization (Figs 4G, S8 and S9). We attribute this result to the need of the cells to properly re-establish their polarity and alignment with the neighbours before the next division and not disrupting the polarity of neighbours while growing. We note that as long as a sufficiently slow growth rate is chosen, the patterning mechanism is very robust, with polarization staying high throughout the whole course of the simulations while the tissue expands, with little variation in global orientation (Fig 4E and 4F).

Our results also indicate a difference between uniform cell proliferation within the tissue, and restricted proliferation at the front, as seen in wound healing, regeneration and PCP establishment in the retina growth cone [59]. For proliferation on front, we observed a shift in the direction of polarity at either $\pm 60$ degrees from the proximal-distal axis that remained coherent for the rest of the simulations. We believe that the shift happens due to the lack of sufficient polarity cues for daughter cells at the front. We noticed a great impact of cell division planes on the onset of this shift, which occurs sooner when cell divisions are restricted along the minor axis compared to divisions restricted along the major axis, as the former division plane leaves the distal-most daughter cell with fewer polarity cues (S6 and S11 Figs). This is further supported by the simulations where cells undergo a single division cycle either inside the tissue or at the front, where we found that cells dividing inside the tissue re-polarize and align their directions ten times faster than cells dividing at the front (S10 Fig). The observed $\pm 60°$ deviation from the proximal-distal axis is surprising as it is not what would be expected due to pinning effects of the lattice. We believe that this angle preference is due to the geometry of our simulation space and the hexagonal packing that arises from front-restricted proliferations, where cells form rows along the direction orthogonal to the proximal-distal axis.

Whether single cells have the ability to polarize cell autonomously seems to be organism specific: in mice, cells are not able to polarize autonomously [52], while in *Drosophila* they can [51]. In our CPM model, we can choose our parameters to model both cell autonomous and cell non-autonomous polarization and our simulations reveal clear differences between both these polarization mechanisms. Although the key results remain qualitatively the same in both autonomous and non-autonomous systems, the latter is better at propagating polarity signals across the entire tissue and establishing long range alignment. This is evident from the results where we observed consistently higher levels of global polarization for non-autonomous systems across all system sizes (S12 Fig). These systems reached 50% global polarization threshold in nearly all simulation runs, and required significantly lesser time to reach that threshold (S13 Fig). Even in a proliferating tissue, the polarization is slightly higher than that observed for autonomous systems ($< 1\%$). Constraints imposed on the affinities between internal cell compartments in the cell autonomous case could be reducing the cell's ability to readjust its polarity direction and possibly letting the system to get trapped in a local minima state. In contrast, the dynamic internal compartments in the non-autonomous systems provide cells with greater flexibility to adapt their polarity to PCP signals from neighbours leading to an optimized tissue alignment. A similar effect was also observed in migratory cells and analysed in more detail by Nandan and Koseska [61,62].

The mechanism we proposed here relies on cell proliferation from an initially small tissue where the low number of cells ensures an initial coherent pattern that can be expanded. However, this raises the question of how global alignment can be established in biological systems where tissues may already be large before PCP activation. We discussed a few alternate mechanisms that may play a role in such cases. One possible mechanism is the use of two signalling boundaries (at the proximal and distal sides) that increases the maximum sizes that pre-growth tissues could polarize coherently. Another mechanism would be a sequence of PCP cell activation from a boundary signal, where the next row of cells only polarizes after the preceding row has already polarized and aligned with the rest of the tissue. A similar mechanism was implemented by Salbreux et al. [59] to model the zebrafish retina development, where tissue scale mechanical stresses also played a role in elongating cells and biasing orientations in a specific direction. Other models also made use of mechanical stresses to aid global PCP alignment. In 2010, Aigouy et al. [63] proposed a vertex model where anisotropic tensions oriented cell elongations and rearrangements, leading to global realignment. A similar mechanism was proposed by Sugimura et al. [64], where anisotropy from cell elongation together with asymmetric cell-cell coupling along a particular axis can act as a global cue for polarity alignment. However, these models are based on a pre-established PCP order from which the applied tension realigns the orientation [63,64], so it is not clear if these mechanisms could by themselves promote alignment from a highly disordered initial state.

There are multiple ways to test whether the mechanism proposed here is at play in any specific tissue. The most direct test is by manipulating the growth rates of cells as the model predicts better(worst) PCP alignments for slower(faster) growth rates. Another model prediction is that the presence of a signalling boundary is necessary to establish the

proximal-distal direction in a growing tissue. If this boundary is removed or made unable to signal/influence the cells, the growing tissue is likely to orient in a different direction. Our model also predicts that the ability of a large tissue to keep its orientation after the removal/blockage of a signalling boundary will depend on cell packing: irregularly packed tissues will experience a slow decrease in their global polarizations proportional to the tissue size, while tissues that are packed in a highly ordered hexagonal tiling would keep their polarization. In the absence of a global cue, our model predicts that a boundary signal alone cannot ensure global polarization of static and large tissues. If PCP is activated in an already large tissue, or blocked until a tissue has grown large enough, the cells would only align locally and global polarization will not be achieved. Our model also suggests that cell non-autonomous PCP polarization provides a more robust polarization mechanism than cell autonomous PCP polarization. This is a species-specific difference, but may be tested by comparing the degree or rate of polarization between tissues of comparable size under similar conditions. Any perturbation that can switch the cell polarization between autonomous and non-autonomous would open the way to test our model predictions about the higher robustness of the cell non-autonomous model.

This work presents the first description of planar cell polarity using the Cellular Potts Model, which offers some advantages and disadvantages compared to the previous models of PCP. Because we are using a phenomenological model, the work here cannot account for the details in molecular pathways nor for the effects of concentrations of different PCP components as done in [38–40]. This shortcoming can be amended by associating each PCP domain with ODE models that capture molecular pathways and signalling, as was done in other CP models [54,65]. Off-grid models, such as cell center and vertex models avoid the undesired effects of lattice anisotropy and allow for an easy implementation of shear stresses on tissues, which would be challenging to do in CP models.

The use of the CPM framework, however, offers some unique advantages by allowing for a detailed representation of cell behaviours in a dynamically evolving tissue environment. Most previous models have considered non-motile cells with fixed shapes [38,39,41,42], with some exceptions incorporating variations in cell shape [43] or allowing cells to divide, change shape, and rearrange their neighbours [63]. In our model, cells are highly dynamic — they can change shape, exchange neighbours, and actively pack themselves through growth and rearrangement. Since PCP is also crucial for providing directional information and thereby guiding other developmental processes, our model based on Cellular Potts formalism provides a versatile framework for studying PCP-driven morphogenesis in other contexts for which Cellular Potts models have been used in the past, such as convergent-extension [66,67], limb-bud growth [68] and primitive streak formation [69]. The mechanisms of long-range polarity without morphogens that we explored here, however, are general and do not rely on our specific choice of modelling framework. We believe that the results of our work should be easy to replicate in other models where cells are able to proliferate and will be interesting to check how other processes such as stress and details on PCP signalling at the molecular level could impact the results.

## Materials and methods

The model was developed using the CompuCell3D (CC3D) simulation software [54], with simulations executed on a computer cluster using a set of custom-developed Python scripts. The simulation code and the analysis scripts are publicly available at https://github.com/abhisha-ramesh/PCP_Subcellular_Potts_Model.

## Supporting information

**S1 Fig. Time taken for global polarization under different scenarios. (A)** An individual cell, in the absence of neighbours can establish PCP asymmetry autonomously and the domains spatially segregate in a timescale of approximately 20 MCS. This timescale is defined as the *polarization time* $(\tau)$ **(B)** Average time for polarization stabilization in 4×4 cell system for periodic boundary and left boundary signal configurations. The average time required for polarization stabilization is determined by binning the time series data and identifying the first interval where the variance in global polarization is

minimal. **(C)** Global polarization vs. time for increasing system size for periodic boundary conditions and the configuration with left boundary signal respectively.
(TIFF)

**S2 Fig. Parameter sensitivity analysis of contact energies for cell autonomous model.** The final global polarization was evaluated for a $4 \times 4$ cell system under periodic boundary conditions after $10^5$ MCS. When varying one contact energy, all other energies and parameters were held fixed as listed in Tables 1, 2, and S1. **(A)** Correlations between the proximal-distal external contact energy ($J_{pd}$) and other external contact energies: proximal-proximal = distal-distal ($J_{pp} = J_{dd}$, left); proximal-lateral = distal-lateral ($J_{lp} = J_{ld}$, middle); and lateral-lateral ($J_{ll}$, right) **(B)** Sensitivity of global polarization to variations in each external contact energy, tested one at a time in the following order: $J_{pd}$, $J_{pp} = J_{dd}$, $J_{lp} = J_{ld}$, $J_{ll}$, $J_{mp} = J_{md}$, and $J_{ml}$. **(C)** Sensitivity of global polarization to variations in internal contact energies: $j_{pd}$ (proximal-distal internal contacts) and $j_{other}$ (all other internal contacts excluding proximal-distal). In all panels, the red box indicates the value used for simulations in the rest of the paper.
(TIFF)

**S3 Fig. (A)** Time evolution of global polarization, quantified by the scalar order parameter $\phi$, under periodic (red) and left-boundary (blue dashed) conditions. In both periodic and left boundary configurations, global polarization increases with increasing time in a system of 8 × 8 cells. The time axis is rescaled in terms of *polarization time* $\tau$. **(B)** Comparison of final global polarization when number of cells in the system is 900 for varying growth rates for front/distal proliferation (blue) and uniform cell proliferation (green). For uniform cell proliferation, the refractory time between cell divisions is set to $10^4$ MCS ($500\tau$). Growth rate is indicated in the units of lattice sites/$\tau$.
(TIFF)

**S4 Fig. Left boundary orienting signal of lateral cell type. (A)** The left boundary orienting signal with the adhesion properties of lateral cell compartment is added. Two of the possible outcomes are shown- all cells point down (left panel), first column of cells points down whereas other columns point up. Global polarization (left panel) and mean tissue angle distribution (right panel) for **(B)** Higher preference of the Medium for proximal or distal compartments over the lateral compartment ($J_{ml} > J_{mp} = J_{md}$). **(C)** Equal preference of the Medium for proximal, distal, and lateral compartments ($J_{ml} = J_{mp} = J_{md}$). Cells do not align along the proximal-distal axis; instead, each column of cells independently chooses between up/down alignment. As a result, the average global polarization decreases, and the mean tissue polarization angle is biased around $\pm 90°$.
(TIFF)

**S5 Fig. Polarization from boundary for cell proliferation for minor axis divisions.** Local polarization decreases with increasing distance from the boundary but stabilizes after approximately 20 columns of cells, for systems with cell proliferation (both uniform cell proliferation and proliferation on front). However, for the system without cell proliferation, the local polarization decreases with increasing distance from the boundary.
(TIFF)

**S6 Fig. Time series data of global polarization, total number of cells and the mean angle of tissue polarization for cell proliferation at the front boundary for (A) minor and (B) major axis cell divisions.** The mean polarization angle is calculated separately for positive and negative values. The mean angle deviates from 0° to $\pm 50°$ as the global polarization starts decreasing.
(TIFF)

**S7 Fig. Time series of polarization misalignment for cell proliferation on front.** Initially, all cells are aligned along the proximal-distal axis. Over time, a defect emerges in one or more cells at the front boundary, which back-propagates

through the tissue, eventually leading to a change in the overall alignment direction for the cells at the front boundary. However, cells near the left boundary signal retain their alignment along the proximal-distal axis.
(TIFF)

**S8 Fig. Comparison of polarization for varying growth rates for cell proliferation (minor axis cell divisions).** Comparison of global polarization vs number of cells for different growth rates under two proliferation scenarios. **(A)** Uniform cell proliferation with cell division along minor axis and refractory time=10,000 MCS. Global polarization decreases with increase in growth rate of cells for all system sizes. There is a decrease in global polarization with increase in the number of cells in the system independent of growth rate. **(B)** Cell proliferation on front with cell division along minor axis. Global polarization decreases with increase in growth rate of cells for all system sizes. Also, there is a decrease in global polarization as the system size increases independent of growth rate.
(TIFF)

**S9 Fig. Comparison of polarization and the mean angle of polarization for uniform cell proliferation (along minor axis) for varying refractory times.** The refractory time ($t_r$) is defined as the maximum interval after a cell divides during which it will begin growing again. Specifically, after a cell divides at time $t_{divide}$, the next growth event is initiated at a time randomly chosen from the interval [$t_{divide}$, $t_r$]. **(A)** Final global polarization and **(B)** mean angle of tissue polarization are compared for varying refractory times ($t_r$) with increasing number of cells in the system. The global polarization remains largely unchanged as the refractory time decreases, except at $t_r = 10^4$ MCS, $10^3$ MCS, and 0 MCS where a slight reduction is observed for all system sizes. The mean polarization angle remains aligned along the proximal-distal axis in all cases except at $t_r = 10^3$ MCS, and 0 MCS where there is a small dispersion about proximal-distal axis. In all cases, cells divide along minor axis.
(TIFF)

**S10 Fig. Time taken by divided cells to re-polarize along proximal-distal axis for different proliferation scenarios.** Starting with a group of 4x9 cells that is already polarized, (1) the first column of cells facing the open boundary is allowed to grow and divide once (2) the fifth column of cells (fully surrounded by other cells) is allowed to grow and divide once. We then calculated the time taken for these cells to polarize along the proximal-distal axis right after they divided. This was done for an ensemble of 100 simulations and the average is plotted below for (1) growth inside and (2) growth on distal front. The results show that cells on the front take much longer to align with the rest of the tissue compared to cells that divide inside the tissue irrespective of the cleavage plane/axis of division.
(TIFF)

**S11 Fig. Effect of the axis/plane of cell division on tissue polarization under different proliferation scenarios.** Comparison of global polarization for cell division along minor axis (green) and major axis (blue) of the dividing cells. **(A)** Illustration of division planes: minor axis corresponds to the short axis of the cell, while major axis corresponds to the long axis. **(B)** Comparison of global polarization versus number of cells for uniform cell proliferation. The refractory time for uniform proliferation is $10^7$ MCS. The global polarization remains unchanged with change in the plane/axis of cell division. **(C)** Comparison of global polarization versus number of cells for proliferation on front boundary. The global polarization changes only by $\approx 1\%$ when the plane/axis of division is changed. The growth rate is 0.001 lattice sites/MCS for both proliferation scenarios. Final tissue angle distributions for **(D)** Uniform cell proliferation for minor (green, left panel) and major (blue, right panel) axis divisions **(E)** Cell proliferation on front for minor (green, left panel) and major (blue, right panel) axis divisions. The angle distribution is primarily along proximal-distal axis for uniform cell proliferation for both minor and major axis cell divisions. However, for proliferation of front scenario, the tissue angle is distributed along $\pm 60°$ for minor axis divisions, whereas the angle is dispersed about the proximal-distal axis for major axis divisions.
(TIFF)

**S12 Fig. Comparison of cell-autonomous vs. cell non-autonomous global polarization.** Panel **(A)** shows the comparison of global polarization for cell-autonomous vs. cell non-autonomous global polarization under periodic boundary conditions with increasing numbers of cells (final state after $10^6$ MCS). Panel **(B)** shows the same comparison under left boundary signal configurations. Panel **(C)** shows the comparison of global polarization for cell-autonomous vs. cell non-autonomous with increasing number of cells for uniform cell proliferation with a refractory time of 10,000 MCS and a growth rate of 0.001 lattice sites/MCS.
(TIFF)

**S13 Fig. Comparison of average time taken to reach** 50% **global polarization for cell-autonomous and cell non-autonomous polarization scenarios. (A) Cell-autonomous polarization** (left): As system size increases along both *x* and *y* directions (square geometry) under periodic boundary and left boundary signal configurations, the time to attain 50% global polarization (solid line) increases, and the fraction of simulations that reach 50% global polarization (dashed line) decreases by approximately 50%. **Cell non-autonomous polarization** (right): As system size increases under the same conditions, the time to reach 50% global polarization (solid line) also increases, but less than in the cell-autonomous case, while the fraction of simulations reaching 50% polarization (dashed line) remains roughly constant near 100%. **(B)** With system size increased along *x* direction while keeping 10 cells constant along *y* direction (rectangular geometry), the time to reach 50% global polarization (solid line) increases with increasing system size for both autonomous and non-autonomous cases, but the increase is higher for cell-autonomous. Although the fraction of simulations reaching 50% polarization (dashed line) decreases with system size for both cases, the decrease is very small for non-autonomous case, whereas it drops sharply to 20% for cell-autonomous case.
(TIFF)

**S14 Fig. Emergence vs maintenance of PCP order.** The maintenance of a pre-established PCP order is studied for a system where **seed** cells fill up the lattice and relax their boundaries. Then, the proximal and distal compartments are allocated to left and right ends of each of seed cells. The maintenance of this polarity is evaluated as time progress until $10^6$ MCS. **(A)** The initial and final polarity vectors for a representative simulation are shown. **(B)** Comparison of global polarization and local polarization for increasing system sizes for systems with pre-established PCP order and systems where PCP is emerging. Global polarization decreases linearly for systems that start with PCP order whereas it increases for systems where PCP order emerges from a random initial configuration. **(C)** As the distance from the left boundary increases, the local order decreases for increasing system size. **(D)** Initial (top panels) and final (bottom panels) tissue angle distributions for increasing system sizes ($10 \times 10$- left panel, blue; $10 \times 30$- middle panel, green; $10 \times 50$- left panel, red). As system size increases, the tissue angle is more dispersed about the proximal-distal axis with increasing time.
(TIFF)

**S15 Fig. Emergence vs maintenance of PCP order for perfect hexagonal packing.** The maintenance of a pre-established PCP order is studied for a system with perfect hexagonal packing for **(A)** horizontal and **(B)** vertical alignment.
(TIFF)

**S16 Fig. Other mechanisms for long range alignment without morphogens (A) Double Boundary Signal.** Schematic of the double boundary signal configuration with complementary signalling boundaries on opposite sides of the tissue. **(B)** Global polarization for the double boundary configuration across system sizes, showing increased polarization (red) compared to the single left boundary case (blue). **(C)** Simulation snapshots of a $10 \times 30$ cell system polarized along the proximal–distal axis by double boundary signals over time. **(D) Cascade of PCP Activation.** Cell columns are sequentially activated for PCP signalling from a local boundary.
(TIFF)

**S1 Table. Simulation parameters.** This table lists all parameters used in the simulations, including volume constraints, neighbour order for lattice site copy events and contact energy calculations and volume fractions of each cell and cell compartments inside the cell.
(PDF)

**S1 Video. Time Evolution of a system of 8×8 cells under periodic boundary conditions.** Time evolution of an 8×8 cell system with periodic boundary conditions over $10^5$ Monte Carlo steps (MCS). Vector fields show the evolution of the cell polarity vectors throughout the simulation. The final global polarization for this simulation is 0.75.
(MP4)

**S2 Video. Time evolution of a system of 30×30 cells under periodic boundary conditions.** Time evolution of a 30×30 cell system with periodic boundary conditions over $10^5$ Monte Carlo steps (MCS). Vector fields show the evolution of the cell polarity vectors throughout the simulation. Domains of local alignment and swirling patterns can be seen. The final global polarization for this simulation is 0.28.
(MP4)

**S3 Video. Time Evolution of a system of 8×8 cells under left boundary signal configuration.** Time evolution of an 8×8 cell system with left boundary signal over $10^5$ Monte Carlo steps (MCS). Vector fields show the evolution of the cell polarity vectors throughout the simulation. All cells align along proximal-distal axis. The final global polarization for this simulation is 0.94.
(MP4)

**S4 Video. Time evolution of a system of 30×30 cells under left boundary signal configuration.** Time evolution of a 30×30 cell system with left boundary signal over $10^5$ Monte Carlo steps (MCS). Vector fields show the evolution of the cell polarity vectors throughout the simulation. Cells align along proximal-distal axis closer to the boundary and alignment is lost after a few columns from the left boundary. The final global polarization for this simulation is 0.35.
(MP4)

**S5 Video. Time evolution of a system under proliferation on front boundary.** Starting from a configuration of 10×4 cells polarized along the proximal–distal axis by a left boundary signal, cells at the front boundary grow and divide. After several rounds of division, alignment along the proximal–distal axis is lost. The simulation runs for a total of $10^6$ Monte Carlo steps (MCS), during which the number of cells increases to 105 and the global polarization reaches 0.84. Vector fields depict the evolution of cell polarity vectors throughout the simulation.
(MP4)

**S6 Video. Time Evolution of a system under uniform proliferation.** Starting from a configuration with 10x4 cells (which polarize along proximal-distal axis with the help of a left boundary signal), all cells in the tissue grow and divide (cells start to grow at random times in the interval $[0,10^5]$ MCS. After a few rounds of division, the alignment along proximal-distal axis is lost. The simulation runs for a total of $10^6$ Monte Carlo steps (MCS), during which the number of cells increases to 522 and the global polarization reaches 0.95. Vector fields show the evolution of the cell polarity vectors throughout the simulation.
(MP4)

## Acknowledgments

We thank Dr. Eric Brooks for his valuable feedback and comments on this work, as well as for informing us about experimental studies on PCP orientation in the absence of morphogens. JMB also thanks Dr. Mansoor Raza for first drawing his attention to research on PCP signaling. We also thank Dr. François Graner and Dr. James Glazier for their feedback on the implementation of certain aspects of the model, and Dr. Juan Manuel Gomez, Dr. Lutz Brusch, Dr. Benedikt Best and

Pedro Cencil Dal Castel for their feedback and comments on this work. We acknowledge the computing resources provided by North Carolina State University High Performance Computing Services Core Facility (RRID:SCR_022168). This work was supported by the North Carolina State University.

## Author contributions

**Conceptualization:** Julio M. Belmonte.

**Data curation:** Abhisha Thayambath.

**Formal analysis:** Abhisha Thayambath.

**Investigation:** Abhisha Thayambath.

**Methodology:** Abhisha Thayambath, Julio M. Belmonte.

**Project administration:** Julio M. Belmonte.

**Software:** Abhisha Thayambath.

**Supervision:** Julio M. Belmonte.

**Validation:** Abhisha Thayambath, Julio M. Belmonte.

**Visualization:** Abhisha Thayambath, Julio M. Belmonte.

**Writing – original draft:** Abhisha Thayambath, Julio M. Belmonte.

**Writing – review & editing:** Abhisha Thayambath, Julio M. Belmonte.

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
