## [Decision Letter · Decision Letter 0]

17 Jul 2025

PCOMPBIOL-D-25-01098

Start Small: A Model for Tissue-wide Planar Cell Polarity without Morphogens

PLOS Computational Biology

Dear Dr. Belmonte,

Thank you very much for submitting your manuscript to PLOS Computational Biology. After careful consideration, the reviewers felt that the manuscript presents an exciting extension of PCP modeling with potentially broad interest but there are several outstanding questions for the manuscript to fully meet PLOS Computational Biology's publication criteria. Therefore, we invite you to submit a revised version of the manuscript that addresses the points raised during the review process. We look forward to your revised submission!

Please submit your revised manuscript within 60 days Sep 16 2025 11:59PM. If you will need more time than this to complete your revisions, please reply to this message or contact the journal office at ploscompbiol@plos.org. Please include the following items when submitting your revised manuscript:

We look forward to receiving your revised manuscript.

Kind regards,

Calina Copos, Ph.D.

Academic Editor

PLOS Computational Biology

Marc Birtwistle

Section Editor

PLOS Computational Biology

**Additional Editor Comments :**

A couple reviewers have pointed out that the simulation code is provided but the simulation output or analysis code are not. Please ensure these are provided in the revised version.

**Journal Requirements:**

At this stage, the following Authors/Authors require contributions: Abhisha Thayambath, and Julio M Belmonte. Please ensure that the full contributions of each author are acknowledged in the "Add/Edit/Remove Authors" section of our submission form.

5) We have noticed that you have uploaded Supporting Information files, but you have not included a list of legends. Please add a full list of legends for your Supporting Information files after the references list.

Potential Copyright Issues:

i) Figure 1. Thank you for stating "Panel A adapted from [17]". Please provide written permission from the copyright holder to publish this under our CC-BY 4.0 license, or remove the figure / replace the image. Please note we do not recommend using standard request forms available on Publishers' websites, as they grant single use rather than republication under an open access license.

7) Thank you for stating "The simulation code ispublicly available at https://github.com/abhisha-ramesh/PCP Subcellular Potts Model." This link reaches a 404 error page. Please amend this to a working link and update your Data Availability Statement in the online submission form accordingly.

**Reviewers' comments:**

Reviewer's Responses to Questions

**Comments to the Authors:**

**Please note that one of the reviews is uploaded as an attachment.**

Reviewer #1: A long-standing question in polarity field is how long-range polarity is established and whether morphogen gradients are required. This paper describes a new computational model of planar cell polarity (PCP) that lacks long-range cues, which have been shown to be required in most other theoretical models of PCP. Using a Cellular Potts framework and modeling the partitioning of proximal-distal compartments using differential adhesion of opposing complexes within and between cells, the authors show that this can give rise to spontaneous polarization with strong local alignment in small tissues. However, larger tissues lack long-range alignment without a directional cue, even when an orientation boundary is provided, giving rise to high local order but little global alignment (swirling patterns), as observed in prior models. Coupling a polarizing boundary in a small field of cells that is allowed to grow with uniform proliferation results in robust large-scale polarization, as daughters cells will repolarize utilizing cues from their neighbors. This work nicely demonstrates that a small tissue adjacent to a polarizing boundary can establish long-range polarization if coupled with cell proliferation, as was previously proposed in vivo in the Drosophila larval wing (Sagner et al, 2012).

Although there are many theoretical models for PCP, the current phenomenological model is distinct in its use of the Cellular Potts framework, and by incorporating growth and proliferation as a mechanism to propagate/expand existing polarization. This is exciting because it points experimentalists searching for the elusive global cue for PCP in a new direction, suggesting that an edge or a boundary, rather than a graded morphogen, may be sufficient especially if polarization is established when the tissue is small. These findings will be of broad interest to those interested in tissue patterning and cell polarity.

I have some comments that would improve the readability of the manuscript.

1. In Figure 2A, the model is described as continuous so that all cells had one junction. But I assume this was not implemented for cases involving a boundary (Fig. 2B) or leading edge expansion. Needs clarification.

2. Given that a ‘distal’ orienting bounding is sufficient to generate aligned polarity in small tissues with growth, I wondered what would happen if the polarization boundary was of ‘lateral compartment’ character. Does this provide any bias to align polarity w/ or w/out cell division? In the mouse skin, polarity initiates from the midline outward (Aw et al 2016).

3. Figure 3 shows the model with leading edge divisions is less successful in propagating polarity compared to uniform divisions. I wondered why this condition was tested in the first place. Are there example of tissues that grow in this way? Perhaps during wound repair or limb outgrowth? Other aspects/assumptions of the model (repolarization during division) are rooted in experimental observations, so it would be helpful to place leading edge divisions in a physiological context as well.

4. Figure S4 plots the minor versus random axis division impact on polarization in the uniform or leading-edge division models. The minor axis had small (~3%) improvement over the random axis. Firstly, the long axis is never explicitly tested and secondly, I think the statement above is too strong for such small improvement.

5. Some of the language related to division orientation is confusing to an experimentalist. Division orientation typically refers to spindle orientation ie, according to Hertwig’s rule, cells divide with the spindle aligned with the cell’s long axis. The cleavage plane is therefore aligned with the short/minor axis. In the discussion, “division along the long axis results in daughter cells maintaining more surface contact with each other than with external neighbours, reducing their access to external polarity cues and leading to weaker polarization.” I think if the authors use the terms ‘spindle alignment’ and ‘cleavage plane’ to describe their data, this will be more accessible to experimentalists.

6. In the discussion, the authors explain why they chose model parameters that allow cell autonomous polarization, but state that other parameters that prevent cell-autonomous polarization are also effective in generating long-range polarization. It would be useful to include these data as a supplementary figure.

7. Figure S1B/C, S5 and TableS1 are not referred to in the text.

Reviewer #2: SUMMARY :

The paper revisits the self-organization problem of Planar cell polarity, PCP, the alignment of cell polarity along epithelial sheets that underlies several morphodynamic processes in vertebrates. Whereas several key molecular pathways have been uncovered over the years and several computational models have shown that local cell-cell interactions can explain self-organization on a local scale, most models to date have required a global cue to explain patterning across large tissues. However, experimental efforts to uncover said global cue have not succeeded so far, raising the question if such a global cue truly exists.

In this work, the authors build a cellular Potts model in which PCP emerges without the need for a global polarity cue, through a combination of local symmetry breaking followed by symmetry maintenance during a proliferative process.

The model is nice because it is the first to show spontaneous emergence of PCP in larger systems without the presence of a global cue, and several interesting analyses are performed throughout the paper. As such, the paper has the potential to be a valuable contribution to the field. However, I do have some open questions that I think should be addressed before publication.

QUESTIONS / COMMENTS :

(1) It was not immediately clear to me what exactly the motivation for some of the experiments and the overall work was. This actually became much clearer when reading the discussion:

1a) for example on P12 : “While the mechanisms of local planar cell polarity …. All these negative results suggest that an alternative mechanism, independent of morphogen gradients, may be required to establish global PCP orientation.” I feel like this captures the essence of the paper quite clearly and concisely – but for me, this message got a bit lost in the introduction, likely because of the amount of detail given about the underlying molecular mechanisms (details which do not seem to be used throughout the paper. My suggestion would be to swap these parts: the discussion paragraph above is ideal for the problem statement in the introduction because it is quite conceptual. The discussion could then go into more detail about how this now ties in with known molecular mechanisms and what it might mean for that field.

1b) The same holds for the motivation for the experiments with different boundary conditions in Fig2, as it was not clear to me what the biological motivation for these experiments was – until the discussion on p12 “Sanger et al proposed that PCP may be established early in development, while the tissue is still small, with the guidance of local boundary signals, and long-range tissue orientation arises through subsequent cell proliferation”. This motivation should ideally provided in the introduction and/or in the results section where these experiments are performed.

1c) The discussion nicely places the results in context of literature. But I would also like to see some ideas on how this model could now be experimentally tested, and how (if at all) it differs in behavior from existing models with global polarity cue. It is nice that this model explains the data without the need for this global polarity cue, but that still does not guarantee that this is the “correct” model. That is fine, but it would be good if the discussion can provide further directions on how to continue from here.

(2) Not all parameter values are described: what is T? What is lambda_V? What is V_t? What is the field size? To ensure the work is reproducible, these details should be added.

(3) P8 “We found that as the number of cells increases, global polarization decreases in both configurations (Fig 3A).” and P10 “This clearly suggests that when the system size increases, there is no long-range order across the entire system in the absence of a directional cue”.

3a) I am not sure this is fair since the time it takes to reach the ordered state also increases (maybe even exponentially?) with system size. From Fig S1C it is clear that the larger systems have not yet plateaued at the maximum simulation time. So from this it is not clear whether it is fundamentally impossible to achieve the global ordering, or that it just takes (much) longer. I think it would be useful to complement this analysis with “time taken until a certain amount of ordering is reached” – e.g. the time it takes to reach phi = 0.5, to give you a sense of how the required time scales with system size.

3b) Related: it would be useful to give some indication of how the CPM time in MCS could be related to real time, if possible. For example, is it known how fast individual cells can reorient, and could you use this to set/estimate this scaling from Fig S1A? And if you do, could you compare the predicted ordering time to some experimental data? (Not perfectly, just in terms of order of magnitude to begin with)? Likewise , is the doubling time then realistic?

(4) The paper occasionally seems to conflate two concepts: emergence of order from disorder, and maintenance of order that is already there. For example P10 “This suggests that the distance over which the polarity alignment is maintained, the correlation length, is very small” and “This indicates that the boundary signal helps maintain the cell polarity”. But these experiments all focus on emergence of the order from an unordered system. Since the time to reach the ordered state also increases rapidly with system size (Fig S1C), this does not allow you to ask the second interesting question: what is the max amount of ordering the system can maintain? For that, I think it would be interesting to do a complementary experiment where the cells are initialized in the ordered state, and you evaluate over what spatial scale this ordering is maintained over time.

(5) P5 “We determined the exact values for each contact energy term by extensive parameter scans aimed at optimizing polarization for small systems.” It would be good if the authors could show the results of that parameter scan. How wide/narrow is the regime where this polarization occurs? Are results in the paper expected to hold across different local polarization parameter scenarios?

(6) P8 “The addition of this boundary signal not only biases the global alignment direction but also enhances the global polarization compared to the periodic boundary configuration for nearly all time steps in the simulation (Figure 2D)”.

6a ) Could this be because the leftmost column of cells has a different shape and (potentially) a less variable, high phi? I think it would be worthwhile to check if this still holds when only considering the middle two columns of cells, which are more comparable in shape between the two scenarios.

6b ) Related: there is of course some anisotropy because this is a lattice-based model, which also has implications for the sides of the (hexagonal) cells. Does the left boundary signal scenario in Fig 2B still give the same 4 outputs as in Fig 2A? To what extent is the distribution of order parameter values (say at the final timestep) similar between the 4 potential output directions shown in Fig 2A? If you put the boundary at an angle, do the results still hold?

(7) P12 “we observe that the alignment is not strictly along the x-axis but instead shifts towards +/- 60 degrees”

7a ) I find this somewhat suspicious because this would correspond to the sides of the hexagonal cell shape. Is there any way to check if this is a “real” result or if anisotropy is at play here? What if you placed the boundary signal at an angle?

7b) Related on p13 “We believe that the directional shift in polarity alignment observed for cell proliferation on the front scenario could be due to the lack of ability of the newly divided cells at the boundary to fully integrate polarity cues from all directions”. Interesting hypothesis, but could you test this further with some exps/further analysis? E.g. testing how cells polarize when surrounded by varying numbers of cells, but you may be able to come up with other experiments as well.

(8) P12 “Results remain the same for slower growth rates, but polarization is gradually lost for faster growth rates, suggesting that cells need some time to adjust their polarity before the next round of division to maintain local tissue alignment (Fig S6)”. Could you further support this hypothesis by comparing the distribution of “time between divisions” to that of “time taken by a single cell to (re)polarize”?

(9) P13 “We chose our model parameters (see Table 2) to reflect the latter case; however, setting the internal contact energies to negative values to prevent cell-autonomous polarization still allows spontaneous tissue polarization, and the key results from our model remain qualitatively the same”. That is nice to hear – but ideally, these results should also be shown in the paper and/or supplement.

MINOR COMMENTS TO THE AUTHOR (no response required):

- P3 “However, all these models are based either on a pre-established PCP order from which the applied tension realigns the orientation or on a specific sequence of PCP activation of cell rows that facilitates global tissue alignment”. What is meant by “sequence of PCP activation of cell rows”? Please clarify.

- Tables 1 and 2: does this imply that the adhesion energies are asymmetric, or do you simply report as a diagonal matrix to avoid duplicate values? Please clarify.

- P3 “Within each cell, these compartments exhibit low affinity for each other, leading to their spatial segregation” – I think in the CPM this would be better described as “actively repel” each other than “have low affinity”, as the latter (neutral adhesion values) would probably not lead to segregation.

- P4 “formed by a collection of lattice sites on a square or hexagonal grid in 2D or 3D. In this work, I assume it is only 2D square? If yes, this should perhaps be mentioned here.

- Fig 1 phi = 0, phi = 1 – I assume these are not the measured values? Consider replacing either with the measured values or phi ~ 0, phi ~ 1 (i.e. replace "=" with "approximate =")

- Fig 1B’ what does the blue box mean?

- Fig 2A what does the red box mean?

- Fig 2C is this the distribution of individual cell angles within a simulation, or of mean cell angles between the 100 simulations?

- Fig 2D can you give an indication of the robustness of this order parameter across simulations? Plot shows the mean, but what is the SD/standard error?

- What boundary conditions were used in Figs 2E/2E’?

- Fig 2A,2B: text refers to “a system of 8 x 8 cells” but the figures show 4 x 4.

Reviewer #3: Attached

**Have the authors made all data and (if applicable) computational code underlying the findings in their manuscript fully available?**

Reviewer #1: None

Reviewer #2: Yes

Reviewer #3: None

PLOS authors have the option to publish the peer review history of their article (what does this mean?). If published, this will include your full peer review and any attached files.

Reviewer #1: No

Reviewer #2: No

Reviewer #3: No

**Figure resubmission:**
---

## [Decision Letter · Decision Letter 1]

26 Nov 2025

PCOMPBIOL-D-25-01098R1

Start Small: A Model for Tissue-wide Planar Cell Polarity without Morphogens

PLOS Computational Biology

Dear Dr. Belmonte,

Thank you for addressing the comments of the three reviewers in great detail. All of them appreciated your work in editing the manuscript. We would like to accept the manuscript but one reviewer asked for two points to be addressed. Therefore, we invite you to submit a revised version addressing these two points for a final editorial review process.

* A rebuttal letter that responds to each point raised by Reviewer #2. You should upload this letter as a separate file labeled 'Response to Reviewers'. This file does not need to include responses to formatting updates and technical items listed in the 'Journal Requirements' section below.

We look forward to receiving your revised manuscript.

Kind regards,

Calina Copos, Ph.D.

Academic Editor

PLOS Computational Biology

Marc Birtwistle

Section Editor

PLOS Computational Biology

**Journal Requirements:**

At this stage, the following Authors/Authors require contributions: Abhisha Thayambath, and Julio M Belmonte. Please ensure that the full contributions of each author are acknowledged in the "Add/Edit/Remove Authors" section of our submission form.

2) We note that your Supplementary Figures files are duplicated on your submission as they are included in supplementary.pdf and SupplFigures.zip files. Please remove any unnecessary or old files from your revision, and make sure that only those relevant to the current version of the manuscript are included.

**Reviewers' comments:**

Reviewer's Responses to Questions

Reviewer #1: The authors have done an excellent job responding to reviewers comments and suggestions. The addition of a new section comparing autonomous vs non-autonomous PCP mechanisms is timely and strengthens the paper. The additional context and model predictions added to the discussion also improve readability. Overall this is a valuable and well executed study that with have substantial impact on the polarity field.

Reviewer #2: I appreciate the care taken in addressing the reviewer comments, which has greatly improved the manuscript.

I have only two remaining (relatively minor) comments:

- It seems as if many of the newly added supplementary figures are only mentioned the first time in the discussion, which reads a bit odd given that the discussion is normally not intended to present new data. I would suggest to integrate these with the results section.

- RE setting the time scale of the CPM: I understand the author's comment that this is difficult because no experimental measurement of polarization time is available, so I'll leave the choice up to them. However, one of the other reviewers mentioned the same issue so I think it might be worth making an attempt. I think that even if polarization time is not experimentally measured, it is probably possible to provide a (very rough!) estimate of at least the order of magnitude of this polarization time. Even if you still measure time in MCS, you could then at least test in what order of magnitude the tissue polarization time would end up , and to what extent that is realistic - I don't think the estimate needs to be perfect to give at least *some* indication that would massively help with interpretation here and as such improve the paper's impact. However, given the large amount of additional work the authors already did, I'll leave the choice up to them (it's not strictly necessary for the paper but would make it stronger if possible).

Reviewer #3: The revision is adequate

**Have the authors made all data and (if applicable) computational code underlying the findings in their manuscript fully available?**

Reviewer #1: Yes

Reviewer #2: Yes

Reviewer #3: None

PLOS authors have the option to publish the peer review history of their article (what does this mean?). If published, this will include your full peer review and any attached files.

Reviewer #1: No

Reviewer #2: No

Reviewer #3: No

**Figure resubmission:**
---

## [Editor Report · Decision Letter 2]

23 Jan 2026

Dear Dr Belmonte,

We are pleased to inform you that your manuscript 'Start Small: A Model for Tissue-wide Planar Cell Polarity without Morphogens' has been provisionally accepted for publication in PLOS Computational Biology.

Best regards,

Calina Copos, Ph.D.

Academic Editor

PLOS Computational Biology

Marc Birtwistle

Section Editor

PLOS Computational Biology

---

## [Editor Report · Acceptance letter]

PCOMPBIOL-D-25-01098R2

Start Small: A Model for Tissue-wide Planar Cell Polarity without Morphogens

Dear Dr Belmonte,

I am pleased to inform you that your manuscript has been formally accepted for publication in PLOS Computational Biology. Your manuscript is now with our production department and you will be notified of the publication date in due course.

With kind regards,

Anita Estes
